# Anisotropic metric-based mesh adaptation for ice flow modelling in Firedrake

Davor Dundovic[1], Joseph G. Wallwork[2], Stephan C. Kramer[3], Fabien Gillet-Chaulet[4], Regine Hock[1,5], and Matthew D. Piggott[3]

[1]Department of Geosciences, University of Oslo, Oslo, Norway
[2]Institute of Computing for Climate Science, University of Cambridge, Cambridge, UK
[3]Department of Earth Science and Engineering, Imperial College London, London, UK
[4]Univ. Grenoble Alpes, CNRS, INRAE, IRD, Grenoble INP, IGE, 38000 Grenoble, France
[5]Geophysical Institute, University of Alaska Fairbanks, Fairbanks, USA

**Correspondence:** Davor Dundovic (davord@uio.no)

**Abstract.**

Glaciological modelling is a computationally challenging task due to its high cost and complexity associated with large spatial- and long time-scale simulations. In this paper we propose feature-based anisotropic mesh adaptation methods and demonstrate their effectiveness for time-dependent glaciological modelling on a Marine Ice Sheet Model Intercomparison Project (MISMIP+) experiment. Our methods use the Python-based Firedrake finite element library and the mmg2d remeshing software. We show that we are able to achieve solution accuracy comparable to a uniform $0.5 \, \mathrm{km}$ resolution mesh simulations by using a sequence of adapted meshes with, on average, 30 times fewer vertices when adapting meshes based on the basal stress and 8–12 times fewer when adapting based on ice thickness and velocity. We further introduce a novel hybrid time-dependent fixed-point mesh adaptation algorithm that reaches mesh convergence approximately twice as fast compared with the existing global fixed-point algorithm. Since the fixed-point algorithms require that the problem is solved multiple times, the reported reduction in number of vertices ultimately translates into a 3–6 times lower overall computational cost compared to uniform mesh simulations.

## 1 Introduction

Despite significant improvements in ice sheet models, the accurate prediction of the contribution of ice sheets to sea-level rise in the 21st century remains uncertain (Aschwanden et al., 2021). A major uncertainty pertains to correctly modelling the dynamics of the grounding line, which is where the ice transitions from being grounded to floating in the ocean (Durand and Pattyn, 2015). Early studies demonstrated that having appropriate refinement of the computational mesh near the grounding line is vital for obtaining reliable numerical results (Schoof, 2007; Vieli and Payne, 2005). This has been further confirmed by the Marine Ice-Sheet Model Intercomparison (MISMIP) experiments, highlighting the mesh resolution's importance regardless of the level of approximation in the force balance equations (Pattyn et al., 2012, 2013). While some studies have proposed methods to mitigate mesh dependency by introducing a smooth transition in the basal stress at the grounding line at the cost

of model accuracy (Leguy et al., 2014), the majority have focused on reducing computational costs by numerical and meshing techniques, while maintaining the model physics. Different strategies have been developed to achieve the latter, including models with explicit mesh movement to solve for the grounding line position (Vieli and Payne, 2005; Moreno-Parada et al., 2023) or fixed-grid models incorporating sub-grid parameterisation of the grounding line (Pattyn et al., 2006; Gladstone et al., 2012; Seroussi et al., 2014). However, these approaches rely on the hydrostatic criterion to define the grounding line position and thus cannot be readily applied to flow models that solve the complete force balance equations and treat the grounding line position as a contact problem (Gagliardini et al., 2016; de Diego et al., 2022).

On the other hand, mesh adaptation methods select finer mesh resolution in regions of interest (e.g., in the vicinity of the grounding line) and coarser elsewhere to preserve numerical resources. These methods have grown particularly popular in finite element and finite volume modelling due to their ability to utilise different types of meshes (structured, unstructured, or hybrid), and, through sophisticated error estimation and adaptation algorithms, to enhance computational efficiency and improve solution accuracy (Alauzet and Loseille, 2016). In the context of mesh adaptation, "solution accuracy" refers specifically to minimising discretisation error, i.e., the error that arises from approximating continuous mathematical equations by discrete numerical methods. Other sources of error are not addressed.

Different criteria have been used in the literature to define the regions of interest with desired mesh resolution. The distance to the grounding line, or alternatively distance from flotation, has been the most popular criterion to define areas warranting smaller mesh sizes (Durand et al., 2009; Goldberg et al., 2009; Gladstone et al., 2010; Cornford et al., 2013; Gudmundsson et al., 2012; Jouvet and Gräser, 2013; Dos Santos et al., 2019). However, such criteria do not explicitly control the solution error and neglect the error contribution of domain regions away from the grounding line, while refining other regions where this may not be necessary, hence making the simulation less efficient. Distance-from-flotation based criteria have been shown to be inferior to refinement criteria based on an error estimator (Dos Santos et al., 2019). Goldberg et al. (2009) uses the jumps in strain rate at cell boundaries as a generic estimator of the numerical error to define areas that require finer mesh resolution. In their method, the total number of mesh nodes is constant and this generic estimator is not adapted to handle all flow regimes as it tends to increase the resolution at the shear margins, sometimes at the expense of the resolution at the grounding line. Dos Santos et al. (2019) implement a true *a posteriori* error approximation (i.e., the actual numerical error is approximated *after* the solution had been obtained), the Zienkiewich-Zhu error estimator, for the deviatoric stress tensor and ice thickness. They find that using the distance to grounding line and the error estimator criteria separately and in combination produce similar results on a benchmark steady-state problem, but predict that the combined criterion would yield superior performance in simulations involving real ice sheets.

Existing ice sheet models utilising mesh adaptation also differ in the way they cope with grounding line movement in time-dependent problems. Some models use fixed meshes, which requires the *a priori* refinement of areas where the grounding line is susceptible to move through, while other models use mesh adaptation techniques that can involve moving meshes (Durand et al., 2009; Goldberg et al., 2009), mesh refinement and/or coarsening (Goldberg et al., 2009; Jouvet and Gräser, 2013; Cornford et al., 2013) or the entire remeshing of the domain (e.g. the Úa finite-element ice-flow model described in

Gudmundsson et al., 2012). However, none of these studies consider the temporal distribution of spatial error associated with time-dependent mesh adaptations.

Mesh adaptation strategies have become popular in computational fluid dynamics to capture complex multiscale phenomena such as shock wave propagation (e.g. Frey and Alauzet, 2005; Alauzet et al., 2007). The general aim is to control the accuracy of the solution by adapting the size and shape (or anisotropy) of individual mesh elements. A core challenge is in finding efficient and reliable 'estimators' of the numerical error which are used as the basis with which to define refinement criteria. Frey and Alauzet (2005) propose an estimator based on the interpolation error; the mesh size is then defined by an anisotropic metric map which equidistributes the error on the computational mesh. Such approaches are also known as *feature-based* approaches, since the constructed metric aims to capture the features of solution fields, or derived solution-dependent fields, from which meshes are to be adapted. While not based on a true a posteriori approximation error, it has been found effective in practice to control the numerical error and offers flexibility in combining metrics obtained for different variables. Generation of anisotropic meshes allows us to capture strongly directional processes and geometries even more efficiently. This is desirable in glacier shear margins, for example, where much finer resolution is required across the shear margin than along it. Such methods have been applied in finite element ice flow models to capture the dynamics of fast-flowing outlet glaciers using a metric defined from the observed surface speed, while keeping overall computational costs low by coarsening other regions (Morlighem et al., 2010; Seddik et al., 2012; Gillet-Chaulet et al., 2012). These methods generate only a single mesh that is used throughout the simulation. However, they do not consider the non-linear and time-dependent coupling between the solution and the underlying mesh. This coupling suggests an iterative mesh adaptation procedure, which we will implement.

The purpose of this paper is twofold. Firstly, we aim to provide a self-contained description of anisotropic metric-based mesh adaptation methods suitable for ice sheet and glacier modelling. Secondly, we demonstrate and evaluate their ability to control solution accuracy while maintaining low computational cost in the context of grounding line dynamics modelling. We build on earlier applications of the method in glaciological modelling and we implement a novel adaptation procedure appropriate for transient simulations, which consists of a spatio-temporal error analysis and a generation of multiple meshes that control the error.

## 2    MISMIP+ experiment setup

The numerical experiment from the third Marine Ice Sheet Model Intercomparison Project (MISMIP+, Asay-Davis et al., 2016; Cornford et al., 2020) features an idealised ice stream, which is an elongated region of fast-flowing ice within an ice sheet. While the ice sheet exhibits negligible basal sliding, the rapid ice stream motion is dominated by processes at the ice-bed interface. Viscous stresses can also be significant in some parts of the ice stream, such as near the grounding line, in shear margins, and where basal traction is low (Greve and Blatter, 2009; Stokes, 2018). The ice experiences a sudden change of flow regime at the grounding line, where the ice stream flows into the floating ice shelf, which is no longer in contact with the bed topography.

We follow the experiment design and prescribed parameter values as described in Asay-Davis et al. (2016), which places the ice stream in an elongated rectangular domain measuring 640 km in the $x$-direction and 80 km in the $y$-direction. The ice is flowing approximately parallel to the $x$-axis over an idealised bedrock topography. The topography is described by a 6th-order polynomial in the $x$-direction and an exponential in the $y$-direction: prescribing an elongated central trough surrounded by steep walls. The bed topography slopes downwards throughout most of the domain, but involves a retrograde slope at around $x = 450$ km where the steady-state ice stream would ground. Beyond the grounding line is the ice shelf that is fed by the upstream flow of ice. The experiment prescribes a no-slip boundary condition at $x = 0$, free-slip conditions at $y = -40$ and $y = 40$ km, and a fixed calving ice front at $x = 640$ km where ice is removed from the domain. The prescribed topography and boundary conditions lead to solutions that are symmetrical about the middle of the domain, at $y = 0$ km. Therefore, we choose to run the experiment in only half of the domain to preserve computational resources, as several participants in the intercomparison have also done (Cornford et al., 2020).

We focus on the Ice1 group of experiments from the MISMIP+ exercise as it produces more drastic glacier evolution and grounding line migration compared to other experiments in the exercise (Cornford et al., 2020). The Ice1 experiment runs for 200 years, where the first 100 years see a retreat of the glacier induced by ice shelf melting. The ice shelf melting is then removed in the last 100 years, when the glacier grows and re-advances. The drastic change of flow regime accompanied with the rapid migration of the grounding line in Ice1 experiment presents an ideal test case for the application of mesh adaptation methods.

## 2.1 Solving equations of glacier flow

Description of the ice stream dynamics requires a momentum conservation equation, which describes how the velocity field $\boldsymbol{u}(\boldsymbol{x}, t)$ evolves under the influence of forces. In this work we apply the Shallow Stream Approximation, which yields the following depth-averaged momentum conservation equation:

$$\nabla \cdot (H\mathbf{S}) + \boldsymbol{\tau}_b = \rho_I g H \nabla s, \tag{1}$$

where $H = H(\boldsymbol{x}, t)$ is the ice thickness, $\mathbf{S} = \mathbf{S}(\boldsymbol{x}, t)$ is the membrane stress tensor, $\boldsymbol{\tau}_b = \boldsymbol{\tau}_b(\boldsymbol{x}, t)$ is basal friction, $\rho_I$ is ice density, $g$ is gravitational acceleration, and $s = s(\boldsymbol{x}, t)$ is surface elevation. The membrane stress and strain rate are related by the Glen's flow power law. Furthermore, we require a description of how ice thickness $H$ evolves in time, which is described by the following depth-averaged mass conservation equation:

$$\frac{\partial H}{\partial t} + \nabla \cdot (H\boldsymbol{u}) = \dot{b}, \tag{2}$$

where $\dot{b} = \dot{b}(\boldsymbol{x}, t)$ is the climatic-basal mass balance rate. For a detailed discussion and derivation of the equations, we refer the reader to Greve and Blatter (2009).

To solve equations of glacier flow we use *icepack*, a Python library built on Firedrake which includes relevant highly customisable glacier flow models (Shapero et al., 2021). As most current ice sheet models, icepack implements the first-order explicit Euler approximation to the coupled model to solve equations of glacier flow. That is to say, at each timestep the

**Table 1.** Number of vertices ($N_v$) and CPU time associated with Ice1 experiment simulations shown in Fig. 2 ran on a single CPU on uniform meshes of varying resolution.

| Resolution (km) | $N_v$ | CPU time |
|---|---|---|
| 4 | 1771 | 2.7 min |
| 2 | 6741 | 9.3 min |
| 1 | 26 281 | 36.8 min |
| 0.5 | 103 761 | 3.34 h |
| 0.25 | 412 321 | 15.85 h |
| 0.125 | 1 643 841 | 83.2 h |

scheme uses ice geometry at the previous timestep to solve the stress balance equations for ice velocity at the current timestep, which is then used to solve the mass transport equation to evolve ice geometry in time (for details, see Shapero et al., 2021).

Icepack formulates the shallow stream model from a principle of least action, where the action functional consists of terms for viscosity, friction, gravity, and terminus. This is a generalisation of the Shallow Shelf Approximation of the Stokes equations as it involves a bed friction term $\boldsymbol{\tau}_b$ which is non-zero for grounded ice and zero for floating ice. Therefore, in order to produce a smooth transition and prevent shocks across the grounding line, we gradually reduce friction in its vicinity (Leguy et al., 2014). To do that, we use a modified form of the Schoof sliding law derived by Shapero et al. (2021). Basal shear stress, $\boldsymbol{\tau}_b$, is then given by

$$\boldsymbol{\tau}_b = \begin{cases} -\dfrac{\alpha^2 N |\boldsymbol{u}|^{1/3}}{\left[(\alpha^2\beta^{-2}N)^4 + |\boldsymbol{u}|^{4/3}\right]^{1/4}} \dfrac{\boldsymbol{u}}{|\boldsymbol{u}|} & \text{if } N > 0 \\ 0 & \text{if } N \leq 0 \end{cases}, \tag{3}$$

where $N$ is the effective pressure at the ice base, $\boldsymbol{u}$ is the horizontal ice velocity, $\alpha^2 = 0.5$, and $\beta^2 = 100 \text{ MPa m}^{-1/3}\,\text{a}^{1/3}$ is the friction coefficient. The effective pressure $N$ is defined as the difference in ice and water overburden pressure, thus $\boldsymbol{\tau}_b$ is zero for floating ice ($N \leq 0$). Finally, icepack implements a Lax-Wendroff scheme for the mass transport model (Shapero et al., 2021). The computational cost for solving the stress balance equations is significantly higher than the mass transport equation.

The weak form of the shallow stream and mass conservation equations are then discretised by Firedrake using finite element methods. To construct an efficient solver for the stress balance equation, we leverage their formulation as the derivative of a convex action functional, meaning that the Hessian is symmetric and positive-definite (Shapero et al., 2021). We therefore solve linear systems using a direct solver based on Cholesky factorisation (Amestoy et al., 2001, 2006). Backtracking line search is used to solve non-linear systems.

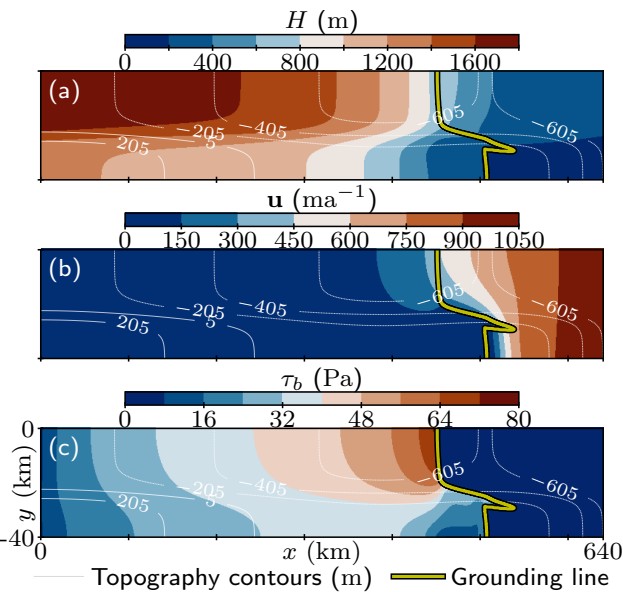

**Figure 1.** Initial steady-state (a) ice thickness $h$, (b) ice velocity $\boldsymbol{u}$, and (c) basal shear stress $\boldsymbol{\tau}_b$ of the MISMIP+ experiment with the sliding law of Eq. (3).

## 2.2 Uniform mesh refinement

In order to study the sensitivity of the solution to mesh resolution and validate the mesh adaptation results, we first run the MISMIP+ Ice1 experiment on a series of uniform structured meshes, with mesh resolutions of 4, 2, 1, 0.5, 0.25, and 0.125 km.

The Ice1 experiment starts from an initial state computed by running the model forward in time for a long time with constant climatic-basal mass balance rate $\dot{b} = 0.3 \ \mathrm{ma}^{-1}$ until the sought quasi-steady state is reached. Following Shapero et al. (2021), we efficiently spin up the model by adopting a hierarchical uniform mesh refinement strategy. We begin with a structured

uniform coarse mesh of 4 km resolution on which we compute a reasonable approximation of the solution. The mesh is then uniformly refined and the simulation continues. This process is repeated five times, resulting in uniform meshes of 4, 2, 1, 0.5, 0.25, and ultimately 0.125 km step-sizes in both $x$- and $y$-directions. The total spin-up time is 15k years. With each mesh refinement, the difference in ice volume and grounding line position at steady state diminishes. We conclude that the simulation has converged at a uniform mesh resolution of 0.25 km, due to very small differences between the 0.5 km, 0.25

km, and 0.125 km resolution results at steady-state. This matches convergence studies performed by MISMIP+ participants, who concluded that the 0.5 km mesh resolution is adequate for the experiments (Cornford et al., 2020, supplementary data). The initial quasi-steady state ice thickness $H$, ice velocity $\boldsymbol{u}$, and basal shear stress $\boldsymbol{\tau}_b$ are shown in Fig. 1, alongside bed topography contours.

Once the steady-state solution had been computed, we initialise the Ice1 experiment by interpolating the steady-state ice

thickness and velocity fields onto uniform resolution meshes. In order to better isolate the differences in results due to different

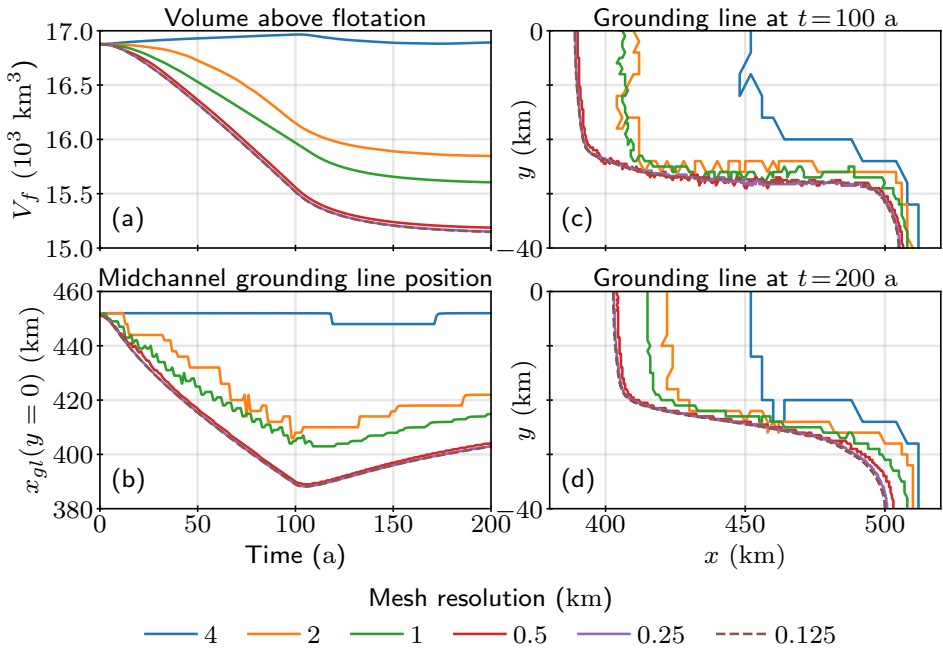

**Figure 2.** Results of the uniform mesh convergence study for the Ice1 experiment for different mesh resolutions: (a) volume above flotation ($V_f$) over time, (b) grounding line position along the midchannel line ($x_{gl}(y=0)$) over time, and (c) grounding line position in the domain at $t = 100$ a and (d) $t = 200$ a.

spatial discretisations, for each simulation we use the same timestep size $\Delta t = 1/24$ $a$ which satisfies the CFL condition on the finest mesh. Results of the Ice1 experiment on each of the uniform meshes are shown in Fig. 2. Finer resolution simulations exhibit faster ice volume loss and retreat of the grounding line, while the coarsest resolution (4 km) finds a new quasi-steady state. There is again a small difference between the 0.125, 0.25, and 0.5 km results in most of the temporal evolution. The largest differences appear in the position of the grounding line along the shear margin (the lateral boundary of the ice stream), where the grounding line position on a 0.5 km mesh fluctuates, and towards the end of the simulation in the grounding line position at the ice ridge near the domain boundary (Fig. 2(d)). Numbers of degrees of freedom and computation times associated with each simulation are given in Table 1, which ranges from 2.7 min for the coarsest resolution, to 83.2 h for the finest resolution simulation. All simulations were run in serial for the purpose of CPU time measurements, but they can be run in parallel.

## 2.3 Mesh adaptation considerations

The convergence study of Fig. 2 confirmed that the MISMIP+ experiment is indeed highly sensitive to mesh resolution. The study, however, does not inform us of what parts of the domain are most sensitive to resolution, since we uniformly refined the mesh over the entire domain. If a certain problem is equally sensitive to resolution throughout the domain, then using a uniform mesh may be the most suitable choice. However, as discussed in detail in the Introduction, earlier studies do show that ice sheet

simulations benefit from selective refinement, particularly near the grounding line (Schoof, 2007; Vieli and Payne, 2005). As general rules of thumb, we would expect that finer resolution is required in regions where the solution changes rapidly, near moving interfaces or free boundaries, around complex geometric features, and so on.

As the name suggests, feature-based mesh adaptation relies on identifying instructive problem-specific features, or *sensors*, to guide mesh adaptation. The choice of a sensor field is non-trivial, as any spatially-varying scalar field qualifies. This includes

the solutions themselves, i.e., the scalar-valued ice thickness and individual components of the vector-valued ice velocity, as well as any solution-dependent field, such as the components of the vector-valued basal shear stress. Multiple sensor fields may also be combined: both in space and in time.

In the next section we show how a metric is defined from a sensor field and how different choices of sensor fields lead to different adapted meshes. For a simpler example, we refer the reader to appendix A, where we apply mesh adaptation methods

to a familiar problem of a Poisson equation.

## 3 Anisotropic metric-based mesh adaptation

Consider a partial differential equation (PDE) problem written in the generic operator form $F(u) = 0$ on a bounded polygonal domain $\Omega$ with an exact solution $u \in V$. Since the domain is bounded, we can prescribe a spatial discretisation in terms of a *mesh*, $\mathcal{H}$, which consists of a finite number of non-overlapping elements and a corresponding number of vertices $N_v$. The

numerical solution $u_h \in V_h$, where $V_h \subset V$ is a discrete subspace of $V$, is obtained here via finite element methods, with an associated approximation error $u - u_h$. The cost of computing the solution $u_h$ on $\mathcal{H}$ increases as $N_v$ increases. However, while increasing $N_v$ through the uniform refinement of a mesh would generally be expected to result in reduced errors, uniform refinement is expected to give a sub-optimal reduction in error as $N_v$ increases. Mesh adaptivity seeks to address this: to more optimally and robustly link a decreasing error with an increasing $N_v$.

The general mesh adaptation problem considered in this paper can be formulated in an a priori way as follows:

$$\text{Find } \mathcal{H}_{\text{opt}} \text{ with } N_v \text{ vertices solving } \min_{\mathcal{H}} \mathbf{E}(\mathcal{H}), \tag{4}$$

where $\mathcal{H}_{\text{opt}}$ is the optimal mesh that minimises some measure of the approximation error $\mathbf{E}(\mathcal{H})$. The goal of mesh adaptation is therefore to achieve minimal approximation error for a given computational cost, which can be directly related to the number of vertices, $N_v$, of $\mathcal{H}$. In anisotropic mesh adaptation approaches presented here, the optimal mesh is found approximately and

iteratively by changing the size, shape, and orientation of its elements.

### 3.1 Continuous metric framework: 2D overview

In order to formulate the mesh adaptation problem (4) such that it is well-posed and that it can be analysed using familiar methods from variational calculus, Loseille and Alauzet (2011a) propose the *continuous metric framework*. The framework establishes a strong duality between discrete mesh elements and continuous mathematical objects stemming from Riemannian

geometry. The key idea in Riemannian geometry is to define a local way of measuring distances in a space, which, unlike

in Euclidean geometry, may vary between points $x \in \Omega$. The notion of measuring distances arises from the definition of a Riemannian metric $\mathbf{M}(x)$, which is identified with a positive-definite matrix in $\mathbb{R}^{d \times d}$ at each point $x$ of the $d$-dimensional domain. A spatially-varying metric leads to local notions of geometric quantities such as length, volume, and angle. This is particularly useful in the context of anisotropic mesh adaptation, as it allows us to prescribe different sizes, orientations, and directional stretchings of individual mesh elements in the domain. We refer the reader to Loseille and Alauzet (2011a, b) for a detailed description and analysis of the framework. However, in order to be self-contained, we illustrate how Riemannian metrics can be used to drive mesh adaptation on a two-dimensional domain $\Omega \subset \mathbb{R}^2$.

It follows from the spectral theorem that $\mathbf{M}$ admits a diagonalisation $\mathbf{M} = \mathbf{R} \operatorname{diag}(\lambda_1, \lambda_2) \mathbf{R}^\mathsf{T}$, where $\mathbf{R} = \mathbf{R}(x)$ is a matrix of orthogonal and normalized eigenvectors of $\mathbf{M}$, and $\lambda_1$, $\lambda_2$ are the corresponding eigenvalues such that $\lambda_1 \geq \lambda_2$. By defining metric density $\rho$ and anisotropy quotients $r_1$, $r_2$ as

$$\rho = \sqrt{\det \mathbf{M}} = \sqrt{\lambda_1 \lambda_2}, \quad r_1 = \sqrt{\frac{\lambda_2}{\lambda_1}} = r_2^{-1}, \tag{5}$$

where anisotropy quotients measure the "stretching" strength in the directions of the eigenvectors in $\mathbf{R}$, the diagonal decomposition can be written in a form that is more instructive in the context of mesh adaptation:

$$\mathbf{M} = \rho \mathbf{R} \begin{pmatrix} r_1^{-1} & 0 \\ 0 & r_2^{-1} \end{pmatrix} \mathbf{R}^\mathsf{T}. \tag{6}$$

In the continuous metric framework, the metric $\mathbf{M}$ prescribes the optimal size of the mesh element (through density $\rho$), its optimal shape (through quotients $r_1$, $r_2$), and its orientation (through eigenvectors in $\mathbf{R}$). Thus, the metric-based approach introduces anisotropy in the resulting mesh.

Analogously to how the metric tensor $\mathbf{M}$ models a single mesh element, the metric tensor field $\mathcal{M} : x \in \Omega \mapsto \mathbf{M}(x)$ models the entire mesh $\mathcal{H}$. To that end, the metric complexity

$$\mathcal{C}(\mathcal{M}) = \int_\Omega \rho(x) \, dx, \tag{7}$$

represents the integrated amount of resolution of $\mathcal{M}$; i.e., it provides an estimate for the number of vertices, $N_v$, in the mesh corresponding to that metric.

As introduced in George et al. (1991), the key idea of anisotropic metric-based mesh adaptation is to generate a mesh which has approximately unit size element edge lengths when viewed in the prescribed Riemannian space. When viewed in the Euclidean space, the mesh is then appropriately adapted and anisotropic. Note that the unity criterion is generally impossible to satisfy exactly (consider discretising a rectangular domain with non-overlapping equilateral triangles in Euclidean space), so the criterion must be relaxed (for details, see Loseille and Alauzet, 2011a).

Following from the diagonalisation of $\mathbf{M}$, the metric is commonly interpreted geometrically by the deformation of a unit circle with a metric map into an ellipse (see Fig. 1.1 in Loseille and Alauzet, 2011a). However, such geometric interpretation is unwieldy for visualising entire metric fields $\mathcal{M}$, and metric density and anisotropy quotient fields may be more appropriate, as shown in Fig. 3. The figure demonstrates how a high metric density $\rho$ prescribes a small adapted mesh element size, and

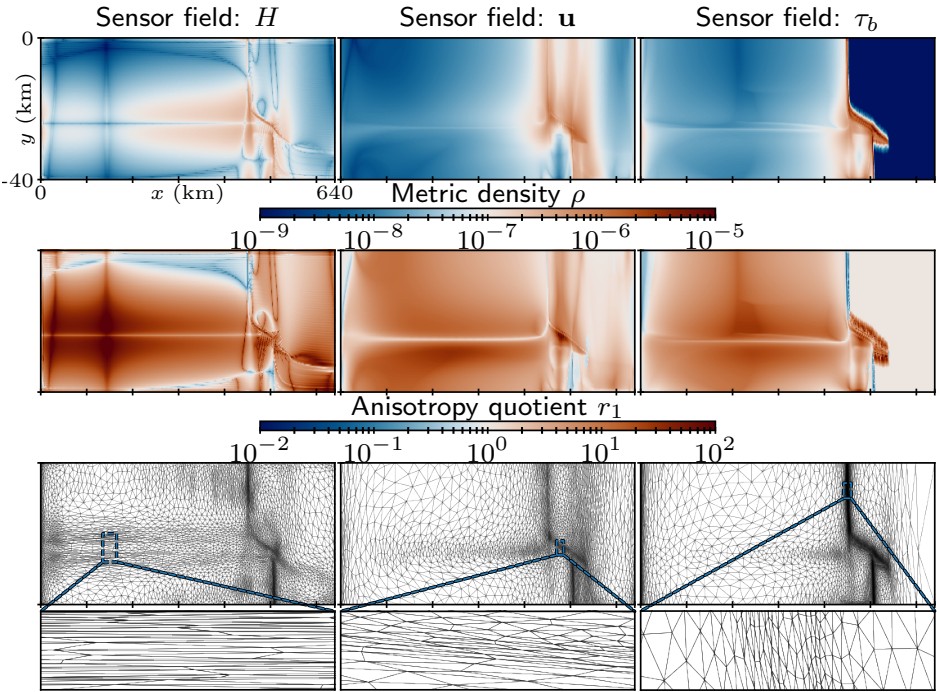

**Figure 3.** Visualising components of metric fields constructed from the Hessian of the initial (left column) ice thickness, (middle column) ice velocity, and (right column) basal stress fields of MISMIP+ experiments (see Fig. 1): (top row) metric density $\rho$ and (middle row) anisotropy quotient $r_1$. The corresponding adapted meshes are shown in the bottom row, including a zoom-in on different regions of the domain. Adjacent adapted edge lengths were not allowed to differ by more than 30%. The zoomed-in subfigures maintain equal scaling of the $x$ and $y$ axes, while the domain is laterally shortened for legibility in other subfigures.

vice versa, while the anisotropy quotients $r_1$, $r_2$ control the stretching of mesh elements in the directions of the eigenvectors in $\mathbf{R}$. For example, the mesh adapted from the basal shear stress metric prescribes finest resolution along the grounding line, where elements' shape and orientation follow the geometry of the grounding line. Furthermore, mesh elements along the shear margin are most strongly stretched along the $x$-direction when adapted from the ice thickness metric.

In cases when the exact solution $u$ is not known, the main challenge lies in finding a reliable estimate of the approximation error $u - u_h$, which is then used in constructing the metric field $\mathcal{M}$. The metric then drives mesh adaptation in a way that reduces the estimated error. In what follows, we consider computing an error estimate based on interpolation error theory.

### 3.2 Error estimate based on the interpolation error

Let us consider the mesh adaptation problem stated in an a priori way in Eq. (4). It can be shown that the approximation error $\|u - u_h\|$ for elliptic problems is bounded by the interpolation error of approximating a continuous field $u$ in $V_h$:

$$\|u - u_h\| \leq c \|u - \Pi_h u\|, \tag{8}$$

where $\Pi_h u$ is the linear interpolant of $u$ onto $V_h$ and $c$ is a mesh-independent constant (Céa's lemma, e.g. Ciarlet, 1991, 2002). In practice, the approach has been found to hold for hyperbolic problems as well (Frey and Alauzet, 2005). Since the solution $u$ and its linear interpolant $\Pi_h u$ are generally not known, the ideas presented in this work are instead applied a posteriori to the numerical solution $u_h$. In the context of general adaptation methods, we are interested in how the error scales with polynomial degree ($p$-adaptation, e.g. Cuzzone et al., 2018; Kirby and Shapero, 2024) and mesh size ($h$-adaptation), which we may readily change. In the former, the polynomial degree can be locally adjusted while maintaining a fixed mesh topology, which eliminates the need for mesh-to-mesh interpolation (see section 3.6). However, higher-order approximations typically lead to increased computational cost and memory consumption. In contrast, modifying the local mesh element size provides greater control over computational efficiency by dynamically adjusting the number of degrees of freedom. In this paper we only consider $h$-adaptation approaches, but both are active and impactful fields of research.

In particular, it can be shown that the local interpolation error $\|u - \Pi_h u\|$ in a single element can be related to the Hessian field of $u$ (see, e.g. Ciarlet, 1991). The result has been readily adopted within metric-based mesh adaptation research due to the (normally) low computational cost involved in computing the Hessian, the directional information contained within it, and the problem-independent nature of Hessian-based interpolation errors (Pain et al., 2001; Frey and Alauzet, 2005; Piggott et al., 2009b; Davies et al., 2011). It has also become a fundamental part of the continuous metric framework, with Loseille and Alauzet (2009) presenting the continuous interpolation error estimation involving a continuous linear interpolant $\pi_{\mathbf{M}} u$. The result was then extended in Alauzet and Olivier (2010), who presented a well-posed continuous formulation of problem (4) that minimises the interpolation error in the $L^p$ norm:

$$\text{Find } \mathcal{M}_{L^p} \text{ which minimises } \mathbf{E}_{L^p}(\mathcal{M}) = \left( \int\limits_{\Omega} (u - \pi_{\mathbf{M}} u)^p \, \mathrm{d}\boldsymbol{x} \right)^{\frac{1}{p}}, \tag{9}$$

under the constraint that $\mathcal{C}(\mathcal{M}) = N_v$.

For two-dimensional steady-state problems, Loseille and Alauzet (2011b) show that the optimisation problem in Eq. (9) has a unique solution which optimally controls the interpolation error in the $L^p$ norm. The resulting optimal metric is given by

$$\mathbf{M}_{L^p} = \mathcal{C}_s \left( \int\limits_{\Omega} (\det |\mathbf{H}|)^{\frac{p}{2p+2}} \, \mathrm{d}\boldsymbol{x} \right)^{-1} (\det |\mathbf{H}|)^{\frac{-1}{2p+2}} |\mathbf{H}|, \tag{10}$$

where $\mathcal{C}_s$ is the target metric complexity and $|\mathbf{H}|$ is the absolute value of the Hessian matrix $\mathbf{H} = \mathbf{H}(\boldsymbol{x})$ of $u$, obtained by taking the absolute value of its eigenvalues in the spectral decomposition. In the numerical demonstrations that follow, we construct the approximation of $\mathbf{H}$ using the robust mixed-$L^2$ projection recovery technique that was heavily influenced by the work of McRae et al. (2018). The approach ensures a continuous representation of second-order derivatives.

The analogous analysis for time-dependent problems was presented in Alauzet and Olivier (2010), which now involves not one, but $N_a \in \mathbb{N}$ meshes $\{\mathcal{H}_i\}_{i=1}^{N_a}$, where each mesh is assigned to a different part of the simulation time interval. They show that the optimal local metric for a subinterval $i$ is given by

$$\mathbf{M}_{L^p}^i = G_{st} \left( n \left( \det |\mathbf{H}_i| \right)^2 \right)^{-\frac{1}{2p+2}} |\mathbf{H}_i|, \tag{11}$$

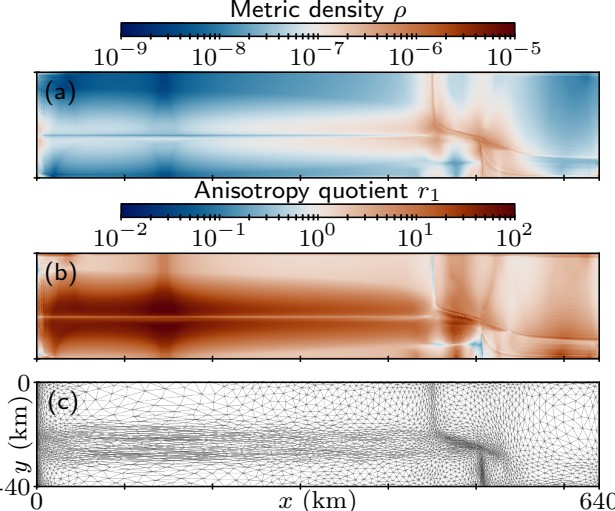

**Figure 4.** (a) Metric density $\rho$ and (b) anisotropy quotient $r_1$ of a metric constructed by intersecting ice thickness and ice velocity metrics shown in Fig. 3, and (c) the corresponding adapted mesh.

where $n$ is the number of timesteps in each subinterval (here constant) and $G_{st}$ is the global space-time normalisation constant

$$G_{st} = \mathcal{C}_{st} \left( n^{\frac{p}{p+1}} \sum_{j=1}^{N_a} \int_\Omega \left( \det |\mathbf{H}_j| \right)^{\frac{p}{2p+2}} \, \mathrm{d}\boldsymbol{x} \right)^{-1}, \tag{12}$$

where $\mathcal{C}_{st}$ is the specified space-time metric complexity. The space-time complexity provides an estimate for the average number of mesh vertices in the entire mesh sequence $\{\mathcal{H}_i\}_{i=1}^{N_a}$. The number of vertices, however, may vary between meshes, as they are distributed in both space and time among individual meshes in the sequence according to Eq. (11) such that the optimal spatio-temporal error in the $L^p$ norm is obtained.

### 3.3 Metric intersection

For many applications, adequately capturing the mesh resolution requirements needs information from multiple sources in the domain. For example, it may be desirable to consider all solutions of a multi-variable PDE of interest, or a vector-valued field whose components may each instruct a different mesh resolution. Furthermore, a time-dependent field may require significantly different mesh resolution in time in order to adequately resolve its evolution. To this end, individual metrics may be constructed from each such sensor field at multiple timesteps, before they are combined to form a single metric that is representative of all those individual timesteps. The combination of metrics is a non-trivial task, and the resulting metric field may vary significantly for different combination approaches and imply different computational cost. Throughout the paper we consider the most common method of combining metrics: metric intersection. An example of an intersected metric is shown in Fig. 4,

where ice thickness and ice velocity metrics of Fig. 3 have been intersected. Inspection of the metric components reveals that the intersected metric indeed contains information from both of the individual metrics.

Metric intersection produces a new metric tensor that captures the most restrictive (i.e., finest) resolution requirements from both original metrics. As described in Alauzet et al. (2007), the process involves transforming the two metrics to a common eigenbasis, taking the maximum eigenvalues from each pair of corresponding eigenvalues, and transforming the metric into the original eigenbasis. That is, given two metrics $\mathbf{M}_1$ and $\mathbf{M}_2$ in two-dimensional space, the intersection $\mathbf{M}_{1\cap2}$ is defined as

$$\mathbf{M}_{1\cap2} = \left(\mathbf{P}^{-1}\right)^{\mathsf{T}} \begin{pmatrix} \max(\lambda_1^1, \lambda_1^2) & \\ & \max(\lambda_2^1, \lambda_2^2) \end{pmatrix} \mathbf{P}^{-1}, \tag{13}$$

where $\mathbf{P}$ is the matrix of of normalised eigenvectors $\{e_i\}_{i=1,2}$ of $\mathbf{N} = \mathbf{M}_1^{-1}\mathbf{M}_2$ and $\lambda_i^j = e_i^{\mathsf{T}}\mathbf{M}_j e_i$. Related to the geometric interpretation of metrics with an ellipsoid (see section 3), metric intersection is visualised as producing a metric whose associated ellipsoid is the largest one that can fit within the ellipsoids of the original metrics (see Fig. 2 in Alauzet et al., 2007).

## 3.4 Metric postprocessing

After the metric had been computed, it may be desired, or even required, to post-process the metric field before the adapted mesh is generated. This includes prescribing the minimum and maximum element edge lengths, as well as maximum tolerated element anisotropy, by appropriately truncating eigenvalues $\lambda_1$, $\lambda_2$ (see Appendix A). Such constraints are particularly useful in order to incorporate existing physical knowledge into the mesh adaptation process, which may not be contained in chosen sensor adaptation fields. Preventing large differences in element sizes and anisotropy between original and adapted meshes may also improve robustness.

Certain choices of sensor fields may lead to metric fields that exhibit a very sharp gradient, as is the case for the metric defined from the basal shear stress, shown in Fig. 3. Borouchaki et al. (1998) show that such large rates of variation may lead to poor quality of adapted meshes. To that end, they introduce a *metric gradation* routine in order to control its variation and therefore smooth out the metric field.

We refer the reader to Dapogny et al. (2014b) for details on the metric gradation routine and edge length truncation implemented in the remeshing library mmg2d, which is used for all mesh adaptation results in this paper (see section 3.7).

## 3.5 Time-dependent mesh adaptation algorithms

Mesh adaptation has been successfully integrated with finite element methods since the 1980s, when the literature almost exclusively focused on applying mesh adaptation to steady simulations (e.g., Peraire et al., 1987). The mesh adaptation procedure involved an iterative process which would typically begin with a coarse uniform mesh. The mesh is then iteratively adapted until the convergence of the steady-state solution is reached, according to a mesh convergence study. In this context, "mesh convergence" refers to the process where successive refinements of the mesh result in progressively smaller changes in the solution, indicating that the solution is approaching a stable state that is independent of further mesh refinement. For details on mesh convergence see, e.g., Loseille and Alauzet (2011a).

Early attempts at mesh adaptation for time-dependent simulations simply employed the above-described algorithm multiple times throughout the simulation. This approach is referred to as the *classical mesh adaptation algorithm* in Alauzet et al. (2007) and we refer to it as such throughout the paper. Namely, the mesh is adapted $N_a \in \mathbb{N}$ times throughout the simulation, at times $t_i$, $i = 0, \ldots, N_a - 1$, based on the current solution state at that time. In this way, the simulation involves $N_a$ different meshes rather than a single fixed mesh, which allows us to prescribe fine resolution in particular regions of the domain *only* at times when that is needed. After the mesh had been adapted, the solution and any other spatially-varying fields in the PDE are then transferred onto it and the simulation resumes until the end simulation time $T$.

While suitable for steady simulations, the classical mesh adaptation algorithm exhibits several shortcomings specific to time-dependent simulations. Firstly, the approach does not control the temporal distribution of error since the metric is only normalised in space according to Eq. (10), as only the current solution is used to guide the mesh adaptation. Additionally, by not considering the future evolution of the solution, the adapted mesh may not adequately resolve the solution at subsequent simulation times. In such a case, the mesh is said to *lag* with respect to the solution. In the context of glaciological modelling, a common consequence of a lagging mesh is the migration of the grounding line out of the fine resolution region of the mesh and into the coarse region, where the grounding line dynamics are no longer accurately captured (e.g., IGE-Elmer/Ice in Sun et al., 2020). While increasing the mesh adaptation frequency would alleviate the lag, doing so may introduce large errors due to frequent solution transfers between meshes (see section 3.6).

To efficiently avoid the mesh lagging behind the solution, the adaptation algorithm requires a prediction of where fine resolution is likely to be required before the next mesh adaptation. An accurate prediction will ensure that the simulated phenomena remain well-resolved when reducing the total number of mesh adaptations $N_a$. Several prediction strategies have been proposed over the years, such as *metric advection* for advection-dominated problems (Wilson, 2010; Smith et al., 2016), and more generally-applicable methods, such as the introduction of safety regions around the fine-resolution region of the mesh (Löhner and Baum, 1992). The latter has become the most common mesh adaptation criterion in marine ice-sheet modelling, where a fine-resolution region was introduced within a set distance from the grounding line. However, such criteria do not explicitly control the solution error and neglect the error contribution of more distant regions, while potentially refining regions where that is not necessary.

Here we adopt the *global fixed-point mesh adaptation algorithm* described in Alauzet and Olivier (2011) in order to remedy the stated challenges specific to time-dependent problems. The algorithm is presented as pseudocode in Algorithm 1. First, the simulation time interval $(0, T]$ is partitioned into $N_a$ equally long subintervals

$$(t_{j-1}, \, t_j] = ((j-1)T/N_a, \, jT/N_a], \tag{14}$$

where $j = 1, \ldots, N_a$ and $T/N_a$ is the fixed subinterval length. To each subinterval we assign a mesh $\mathcal{H}_j^{(k)}$, where $k$ is the fixed-point iteration number. At each iteration $k$, the PDE problem is solved over the entire interval $(0, T]$ on the sequence of meshes $\{\mathcal{H}_j^{(k)}\}_{j=1}^{N_a}$. From the obtained solutions we compute Hessian metrics for given discrete timesteps and combine them in time according to Eq. (13). The computed metric fields are then space-time normalised according to Eq. (11). In such a way we obtain the metric field $\mathcal{M}_j^{(k)}$ associated with each subinterval $(t_{j-1}, \, t_j]$. Finally, the meshes are adapted to yield the next

**Algorithm 1** Global fixed-point mesh adaptation algorithm.

---

**for** iteration $k = 1, \ldots, k_{\max}$ **do**

    **for** time subinterval $j = 1, \ldots, N_a$ **do**

        Interpolate the initial condition/PDE solution onto $\mathcal{H}_j^{(k)}$

        Solve PDE on subinterval $(t_{j-1}, t_j]$

        Construct subinterval metric $\mathcal{M}_j^{(k)}$

    **end for**

    Normalise subinterval metrics $\{\mathcal{M}_j^{(k)}\}_{j=1}^{N_a}$ in space and time

    Generate adapted meshes $\{\mathcal{H}_j^{(k+1)}\}_{j=1}^{N_a}$

    **if** converged **then**

        Terminate

    **end if**

**end for**

---

iteration's adapted mesh sequence $\{\mathcal{H}_j^{(k+1)}\}_{j=1}^{N_a}$ and the simulation is restarted. The algorithm terminates when a set maximum number of iterations or prescribed convergence criteria have been reached.

The global fixed-point adaptation algorithm offers several advantages over the classical mesh adaptation algorithm. Firstly, the algorithm remedies the lagging mesh problem, since the problem is first solved over the entire time interval. The constructed metric fields therefore contain a prediction of the solution evolution over the entire subinterval. Secondly, metric fields are normalised according to space-time normalisation in Eq. (11), which allows for the error to be controlled in both space and time. On the other hand, since meshes are adapted at the end of each iteration, each subinterval may start with a worse approximation of the initial state compared to the classical mesh adaptation algorithm, which adapts the subinterval mesh before the PDE is solved over the corresponding subinterval. This may imply significant additional computational cost if the initial mesh sequence does not accurately capture the modelled phenomena, which is often the case for coarse uniform meshes. As a result, mesh convergence may be slow and many fixed-point iterations may be required.

Here we propose a combination of the classical and global fixed-point mesh adaptation algorithms, which yields a better initial approximation of the optimal mesh sequence at a minimal additional cost. We achieve this by incorporating the classical algorithm within the first iteration of the global fixed-point algorithm. The latter iterations then proceed as in the global fixed-point iteration, without incorporating the classical mesh adaptation algorithm since the mesh sequence is already well approximated. The proposed hybrid algorithm is presented in Algorithm 2. Since the classical adaptation algorithm generates the adapted mesh sequence $\{\mathcal{H}_j^{(1)}\}_{j=1}^{N_a}$ before the PDE is solved on each subinterval, the only added computational cost compared to the global fixed-point algorithm is that of adapting the meshes. This added cost is normally small compared to the cost of solving the PDE. Solution fields obtained from the first iteration of the hybrid algorithm are therefore more accurate than those from the global fixed-point algorithm.

**Algorithm 2** Hybrid fixed-point mesh adaptation algorithm.

---

**for** iteration $k = 1, \ldots, k_{\max}$ **do**
    **for** time subinterval $j = 1, \ldots, N_a$ **do**
        **if** $k = 1$ **then**
            Construct $\mathcal{M}$ from $u(\boldsymbol{x}, t_{j-1})$
            Normalise $\mathcal{M}$ in space
            Generate adapted mesh $\mathcal{H}_j^{(k)}$ from $\mathcal{M}$
        **end if**
        Interpolate the initial condition/PDE solution onto $\mathcal{H}_j^{(k)}$
        Solve PDE on subinterval $(t_{j-1}, t_j]$
        Construct subinterval metric $\mathcal{M}_j^{(k)}$
    **end for**
    Normalise subinterval metrics $\{\mathcal{M}_j^{(k)}\}_{j=1}^{N_a}$ in space and time
    Generate adapted meshes $\{\mathcal{H}_j^{(k+1)}\}_{j=1}^{N_a}$
    **if** converged **then**
        Terminate
    **end if**
  **end for**

---

## 3.6 Interpolation between meshes

The final step of any mesh adaptation procedure is that of interpolating all spatially varying fields to the newly generated mesh. This is a crucial step in the mesh adaptation process, as the choice of transfer method may significantly affect the accuracy of the solution. Since the optimal choice of the transfer method largely depends on the specific problem at hand, we discuss it here with respect to glaciological applications. The process also depends on the type of mesh and mesh adaptivity used. For example, Cornford et al. (2013) use the more restrictive block-structured meshes in order to alleviate issues related to numerical diffusion and mass conservation that we discuss below for unstructured meshes.

The mesh adaptation toolkit Animate, described in section 3.7, currently offers three methods for interpolating fields between different meshes: linear interpolation, Galerkin projection, and the bounded variant of the Galerkin projection (see Farrell, 2010, for an overview of these methods). The most popularly used interpolation method in the majority of applications is that of linear interpolation. However, linear interpolation does not preserve the integral of interpolated fields and its application to ice flow problems would yield unphysical results due to conservation laws being violated. The use of Galerkin projection is therefore more appropriate since it is a conservative operator, despite implying a larger computational cost. In particular, Galerkin projection may introduce new function extrema, which may be problematic in times of extreme glacier retreat when ice thickness is close to zero. In such cases, the interpolated ice thickness might become negative-valued. In order to prevent that, we use the bounded variant of the Galerkin projection for degree-1 basis ($P_1$) fields, which prevents the addition of

new function extrema (Farrell et al., 2009; Farrell and Maddison, 2011). However, by bounding the interpolated solution, the operation introduces additional numerical diffusion which may diminish the accuracy of the overall simulation.

## 3.7 Software

Firedrake (Ham et al., 2023) is a Python-based finite element library for the numerical solution of PDEs utilised throughout this paper. It provides a high-level interface for the definition of finite element variational forms using the Unified Form Language (UFL, Alnæs et al., 2014) which are then compiled into efficient low-level code for the solution of the PDEs. Firedrake utilises PETSc for solving linear and non-linear systems, as well as for its underlying mesh concept (Balay et al., 1997, 2023; Lange et al., 2016).

Mesh generation in Firedrake can be performed using built-in classes, reading in meshes generated by external generators (such as gmsh, Geuzaine and Remacle, 2009), or by using Netgen (Schöberl, 1997), an external generator that has been integrated into Firedrake. Recent developments involve the *Animate* library for metric-based anisotropic mesh adaptation (Wallwork et al., 2024a). Animate implements the presented Riemannian metric framework, Hessian recovery methods, as well as metric operations and interfacing with PETSc's Riemannian metric functionality (Wallwork et al., 2022). A separate library, *Goalie*, has been developed for time-dependent mesh adaptation and supports both feature- and goal-oriented adaptation approaches (Wallwork et al., 2024b). Both libraries have undergone development synchronously with the development of the work presented in this paper.

The process of locally adapting meshes with respect to metric fields is performed by the external library Mmg2d, which is part of the Mmg platform (Balarac et al., 2022; Dobrzynski and Frey, 2008) that has been integrated into PETSc. Mmg2d requires mesh topology and a metric field defined at each mesh node as input, and returns the mesh where elements' size, shape, and orientation have been adapted to the given metric field. While the work presented in this paper focuses on 2D mesh adaptation, Animate and Goalie also support 3D mesh adaptation through the implementation of the Mmg3d and ParMmg mesh generation toolkits (Dapogny et al., 2014a).

## 4 Results

In this section we demonstrate the ability of a Hessian-based anisotropic mesh adaptation method to control the solution accuracy in the MISMIP+ Ice1 experiment. We consider constructing metric fields from the Hessian of ice thickness $H$ and ice velocity $\boldsymbol{u}$, their intersection, and the Hessian of basal stress $\boldsymbol{\tau}_b$. The basal stress field is computed from Eq. (3) and the hydrostatic approximation for the normal basal stress, and as such depends on both $H$ and $\boldsymbol{u}$.

In the absence of a closed-form solution, we rely on the 0.25 km resolution results to define reference solutions $\boldsymbol{u}_{\text{ref}}(\boldsymbol{x},t)$ and $H_{\text{ref}}(\boldsymbol{x},t)$. We validate the results of each simulation by computing the relative solution error in $L^2$ norm based on the

reference solutions:

$$\tilde{e}_H = \frac{\|H_{\text{int}} - H_{\text{ref}}\|_{L^2}}{\|H_{\text{ref}}\|_{L^2}}, \tag{15}$$

$$\tilde{e}_{\boldsymbol{u}} = \frac{\||\boldsymbol{u}_{\text{int}}| - |\boldsymbol{u}_{\text{ref}}|\|_{L^2}}{\||\boldsymbol{u}_{\text{ref}}|\|_{L^2}}, \tag{16}$$

where $H_{\text{int}}$ and $\boldsymbol{u}_{\text{int}}$ are solution fields interpolated onto the reference 0.25 km mesh and $|\boldsymbol{u}|$ is the velocity magnitude. We further consider the maximum deviation of the volume above flotation

$$\|\Delta V_f\|_\infty = \max_i |V_f(t_i) - V_{f,\text{ref}}(t_i)|, \tag{17}$$

and midchannel grounding line position

$$\|\Delta x_{gl}\|_\infty = \max_i |x_{gl}(y=0, t_i) - x_{gl,\text{ref}}(y=0, t_i)| \tag{18}$$

over all simulation years $t_i$, where $V_{f,\text{ref}}$ and $x_{gl,\text{ref}}(y=0)$ are the ice volume above flotation and grounding line position along the midchannel line, respectively, for the reference 0.25 km uniform mesh simulation (see Fig. 2).

## 4.1 Mesh adaptation strategy

In our mesh adaptation experiments we do not incorporate the gained insight into resolution sensitivity from section 2.2. As such, we do not prescribe bounds on element edge lengths and element anisotropy. We use Mmg2d's default maximum metric gradation factor of 1.3, which prevents adjacent element edge lengths to differ more than 30%.

Crucially, when solving time-dependent mesh adaptation problems, we must decide how to split the time domain into $N_a$ distinct subintervals. As discussed in section 3.5, the choice should consider the balance between accurately resolving the ice stream in time, and the added interpolation error and computational cost associated with each mesh adaptation. To that end, we choose to split the simulation time interval $(0, 200\,\text{a}]$ into 20 equally long subintervals, meaning that meshes are adapted every 10 years of simulation time. A constant timestep $\Delta t = 1/24\,\text{a}$ is again used to ensure that the CFL condition is satisfied.

Similarly, we choose to consider solutions at every 1 year of simulation time in the construction of the metric fields. This provides us with 10 metric fields per subinterval, which is, again, a compromise between accurately representing the evolution of the ice stream and avoiding further computational cost associated with metric computations. The 10 metric fields are combined in time using metric intersection of Eq. (13) to yield a single subinterval metric. This is repeated for each of the 20 subintervals. Afterwards, the 20 subinterval metrics are normalised in space and time according to Eq. (11).

It must be noted that there are certainly other viable choices of partitioning the time interval, which we do not explore here. For example, considering that the thinning of the ice shelf and grounding line retreat is more extreme in the first half of the experiment (Fig. 2), a reasonable choice might involve more frequent adaptations in the first half of the time interval than in the second half. However, we do not incorporate such prior knowledge in the setup of the experiments, and rely on space-time metric normalisation to distribute available degrees of freedom accordingly. Similarly, we always prescribe an equal target complexity to each subinterval metric in the first iteration of the hybrid fixed-point algorithm, which cannot utilise

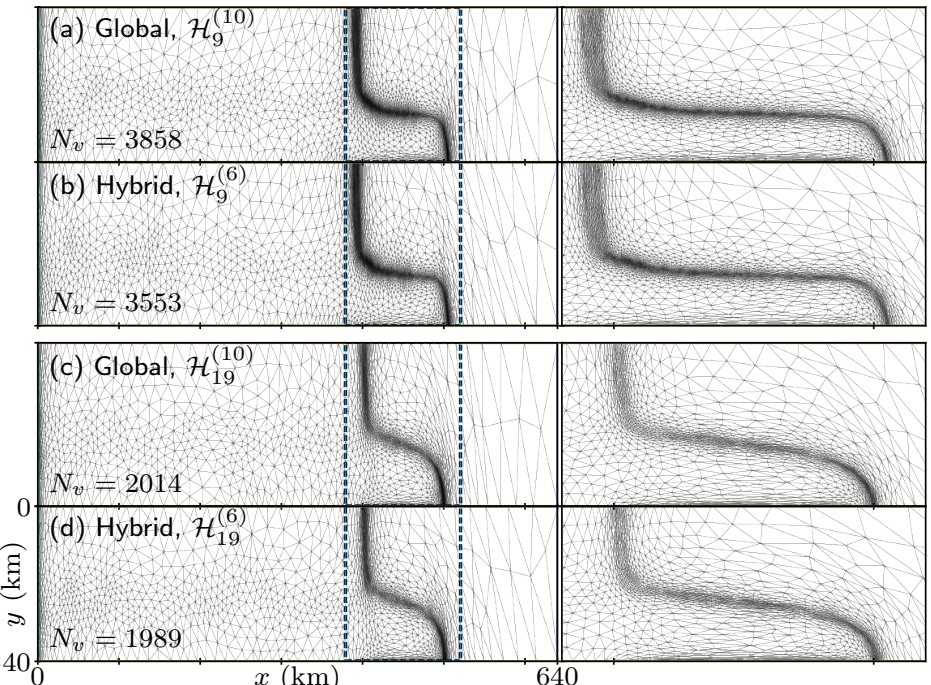

**Figure 5.** Comparing final adapted meshes obtained from the global fixed-point and hybrid mesh adaptation algorithms for target complexity $C = 1600$. Final adapted meshes $\mathcal{H}_9$ corresponding to the (90 a, 100 a] subinterval are shown in (a) and (b), and final subinterval meshes $\mathcal{H}_{19}$ corresponding to the (190 a, 200 a] subinterval are shown in (c) and (d), with a zoom-in around the grounding line on the right. Meshes adapted using the global algorithm are shown in (a) and (c), while meshes from the hybrid algorithm are shown in (b) and (d).

space-time normalisation. This will result in the numbers of vertices, $N_v$, of each mesh of the mesh sequence $\{\mathcal{H}_j^{(1)}\}_{j=1}^{N_a}$ to be approximately equal.

For interpolating fields between meshes, we use the bounded variant of the Galerkin projection. As discussed in section 3.6, its use is most appropriate in the experiment considered here as it is a conservative operation that does not introduce new extrema in the solution. Since the bounded variant of the Galerkin projection is only implemented for $P_1$ fields, we use $P_1$

functions to represent both ice velocity and ice thickness over elements. The leading-order error for this discretisation typically scales quadratically with the decreasing mesh size, in the $L^2$ norm. However, the reduced regularity due to a discontinuity in the derivative of $\boldsymbol{\tau}_b$ in practice results in a degradation of the convergence rate.

### 4.2 Comparing time-dependent mesh adaptation algorithms

We first wish to compare the mesh convergence rate of the global and hybrid fixed-point adaptation algorithms described in

section 3.5. To that end, we run experiments for the two approaches. The first iteration of the global fixed-point algorithm uses an initial mesh sequence $\{\mathcal{H}_j^{(1)}\}_{j=1}^{N_a}$ of $N_a = 20$ uniform 4 km resolution meshes, while the hybrid algorithm begins with meshes generated using the classical mesh adaptation algorithm. Meshes are adapted based on the Hessian of the basal stress

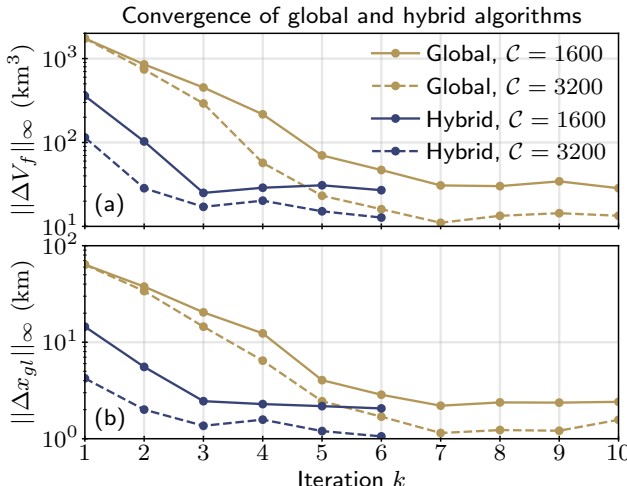

**Figure 6.** Comparing solution convergence of global and hybrid mesh adaptation algorithms, with basal stress as the adaptation sensor field and target complexities $\mathcal{C}$ of 1600 and 3200. The resulting maximum deviation from the reference solutions of the (a) volume above flotation, $\|\Delta V_f\|_\infty$, and (b) midchannel grounding line position, $\|\Delta x_{gl}(y=0)\|_\infty$, over all simulation years are shown.

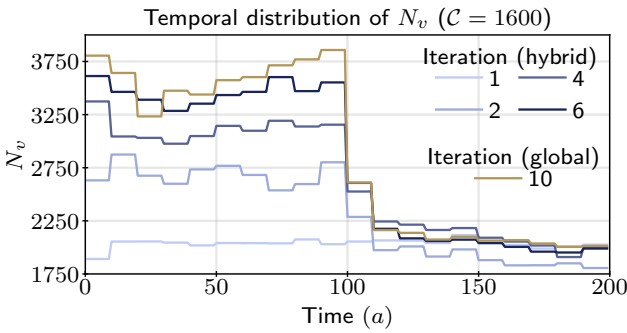

**Figure 7.** Temporal distribution of number of vertices, $N_v$, for several iterations of the hybrid algorithm and the final iteration of the global fixed-point algorithm for $\mathcal{C} = 1600$.

$\tau_b$, but the same conclusions follow from other choices of adaptation sensor field, as discussed below. We repeat the procedure for two choices of target complexities $\mathcal{C}$: 1600, which is comparable to the number of vertices of the uniform 4 km resolution

mesh (see Table 1), and 3200.

As shown in Fig. 5, the global and hybrid mesh adaptation algorithms converge to qualitatively similar adapted meshes. However, the hybrid algorithm converges to a reasonable approximation of the optimal mesh twice as fast, in only 3 iterations, while the global algorithm converges in 7 iterations, as shown in Fig. 6.

In the first iteration of the global fixed-point algorithm, the initial mesh sequence of uniform 4 km resolution is not able to

470 adequately resolve the ice stream dynamics, as shown previously in section 2.2. The poor solution accuracy in the first iteration

in turn leads to suboptimal metric fields and adapted mesh sequence used in the next iteration. This result is independent of the choice of sensor field or target complexity used in constructing the metric, since it is the initial uniform mesh sequence that is far from optimal. In comparison, deviations in the initial iteration of the hybrid fixed-point algorithm are up to an order of magnitude smaller since the ice stream is not modelled on the inadequate coarse uniform mesh sequence.

Furthermore, the two algorithms converge to adapted meshes with consistent vertex counts, as shown in Fig. 5 and Fig. 7. However, the meshes adapted in the first iteration of the hybrid algorithm are only normalised in space (see Algorithm 2), meaning that the number of vertices of each mesh in the mesh sequence $\{\mathcal{H}_j^{(1)}\}_{j=1}^{N_a}$ is therefore approximately equal. As in the global fixed-point algorithm, the metric fields used to generate the next mesh sequence $\{\mathcal{H}_j^{(2)}\}_{j=1}^{N_a}$ are constructed from solutions sampled from the entire time domain, so the metric fields are able to be normalised in time as well as in space. The adapted mesh sequences $\{\mathcal{H}_j^{(k)}\}_{j=1}^{N_a}$, $k \geq 2$ now contain a higher number of vertices in the first half of the simulation, and fewer in the second half. This is expected since the grounding line migration is more pronounced in the first half of the simulation, and requires a more widely refined mesh in order for the grounding line to remain within the finely refined mesh region as it migrates in time.

Overall, we have found that a reasonable mesh convergence may be expected in only a few iterations of the hybrid fixed-point algorithm. Namely, most of the effort is done in the first two iterations of the algorithm. Given a reasonable time interval partition and sufficient target complexity, the classical mesh adaptation algorithm embedded within the first iteration generates an adequate adapted mesh sequence $\{\mathcal{H}_j^{(1)}\}_{j=1}^{N_a}$. The mesh sequence $\{\mathcal{H}_j^{(2)}\}_{j=1}^{N_a}$ is then generated at the end of the first iteration from metric fields normalised in both space and time, which allocates additional resolution to subintervals where that is necessary. Solutions obtained in the second iteration are therefore even more accurate, as the error is optimally controlled in space and distributed in time. The mesh sequence generated at the end of the second iteration, $\{\mathcal{H}_j^{(3)}\}_{j=1}^{N_a}$, can therefore be assumed to be a reasonable approximation of the optimal mesh sequence.

### 4.3 Comparing adaptation sensor fields

Finally, we study the mesh convergence and solution accuracy in simulations utilising the hybrid fixed-point adaptation algorithm for different adaptation sensor fields. We repeat the mesh adaptation process for 3 global-fixed point iterations, which implies a total of 4 adaptations of each mesh. As we demonstrated in subsection 4.2, this is enough to achieve reasonable mesh convergence. For each choice of sensor field, we prescribe the target metric complexity $\mathcal{C}$ of 800, and repeat the simulation 4 more times such that the target complexity is doubled each time. The final simulation therefore specified the target complexity of $16\mathcal{C} = 12800$. Adapted meshes $\mathcal{H}_9$ are shown in Fig. 8 for simulations specifying target complexity of $8\mathcal{C}$, with their elements' aspect ratios shown in Fig. 9. The spatial error distribution at $t = 100$ a on $\mathcal{H}_9$ for $8\mathcal{C}$ is shown in Fig. 10, while the time-averaged errors are shown in Fig. 11

Figure 8 reveals instructive areas of refinement and coarsening for each constructed metric field. When adapting based on ice thickness, the fine resolution is concentrated in two regions: along the shear margin and on the ice shelf near the grounding line. This means that the grounding line will easily migrate out of the fine resolution region during glacier retreat, as shown by the grounding line contours of the reference solution in Fig. 8(a). On the other hand, the grounding line will remain in the

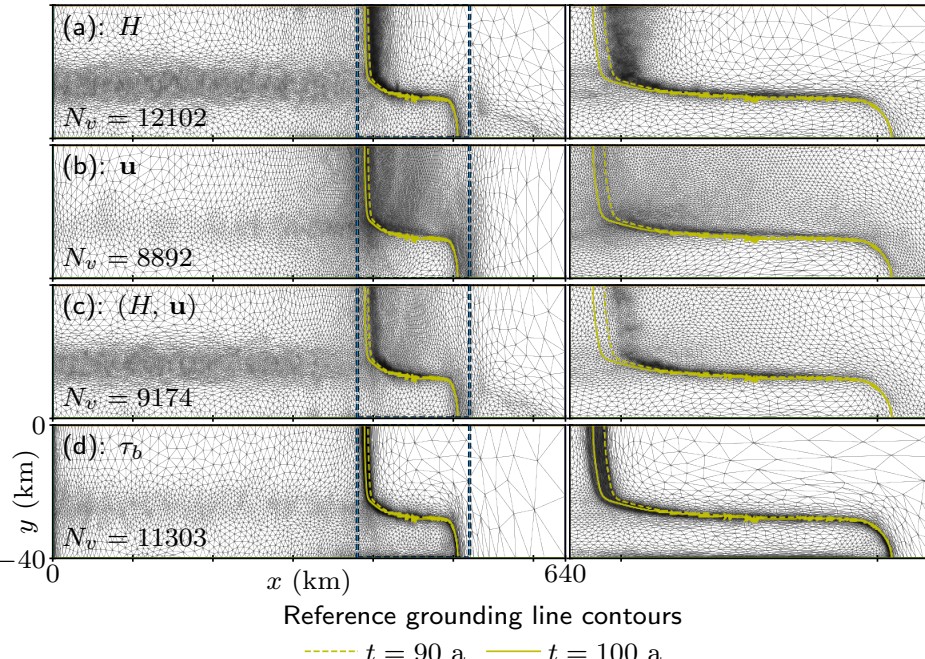

**Figure 8.** Adapted meshes $\mathcal{H}_9$ associated with the $(90\,\text{a}, 100\,\text{a}]$ subinterval, with a zoom-in around the grounding line on the right. Metrics were constructed from (a) $H$, (b) $\boldsymbol{u}$, (c) intersected metric of $H$ and $\boldsymbol{u}$, and (d) $\boldsymbol{\tau}_b$. Shown grounding line contours are computed at the beginning and end of the subinterval from the reference $0.25\,\text{km}$ uniform mesh resolution simulation solutions.

fine region during glacier advance, as it migrates further downstream. In comparison, meshes adapted based on ice velocity are mostly refined in the vicinity of the grounding line and in the inner part of the ice shelf, where ice quickly speeds up as it is no longer influenced by friction at the bed. However, the region around the grounding line is not as finely-resolved as in Fig. 8(a); instead, the available resolution is more-or-less equally distributed over a much larger region. As expected, meshes adapted from the intersected metric of the two fields exhibit features of both. However, this also includes a less refined region upstream of the grounding line compared to that in Fig. 8(b), which again leads to the migration of the grounding line into the coarser region of the mesh. Meanwhile, because the basal stress rapidly diminishes across the grounding line, the metric field computed from its Hessian will prescribe the finest mesh resolution in its vicinity, as shown in Fig. 8(d). This ensures that the grounding line remains in the finely-resolved region of the mesh throughout the subinterval. However, since basal stress is null in the ice shelf, the region of the mesh corresponding to it is coarsened. Variation in mesh cell sizes in the ice shelf is only due to the metric gradation routine.

A measure of cell aspect ratios is shown in Fig. 9, where a unit aspect ratio corresponds to a perfectly isotropic cell (i.e., an equilateral triangle). We observe that meshes whose metrics involve ice thickness contain two bands of highly anisotropic cells in the shear margin with a predominantly $x$-directional orientation separated by a band of isotropic cells. Since the variation of ice thickness is much sharper across the shear margin than along it, anisotropic elements help to efficiently capture such

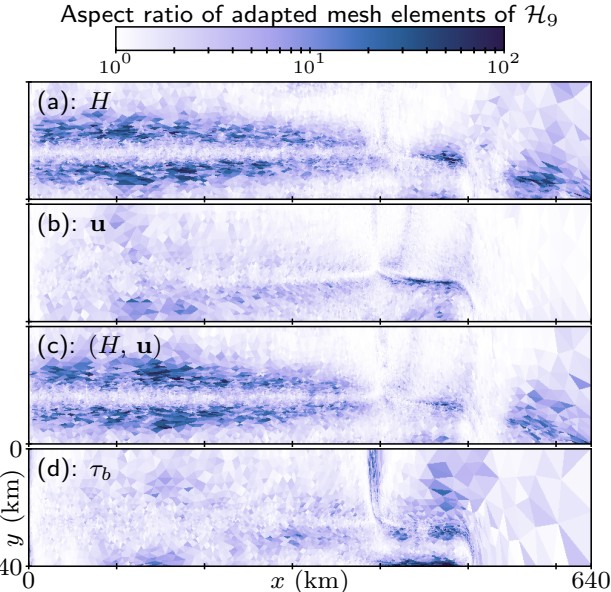

**Figure 9.** Aspect ratio of elements of adapted meshes $\mathcal{H}_9$ shown in Fig. 8, corresponding to the time subinterval $(90\,\mathrm{a}, 100\,\mathrm{a}]$. The meshes are adapted from the Hessian of (a) $H$, (b) $\boldsymbol{u}$, (c) intersected metric of $H$ and $\boldsymbol{u}$, and (d) $\boldsymbol{\tau}_b$.

directional processes along the shear margin by avoiding unnecessary over-refinement in the $x$-direction. Similarly, in order to capture the rapid variation of the basal stress across the grounding line, meshes adapted based on the basal stress have elements stretched in the direction of the grounding line along its entirety. On the other hand, meshes adapted from ice velocity contain mesh elements with lower anisotropy along the ice stream shear margin, but strongly anisotropic elements in the $x$-direction along the grounding line flank as the ice drastically speeds up across it.

The above observations directly reflect in the spatial distribution of the solution errors, shown at $t = 100\,\mathrm{a}$ in Fig. 10. While the error in $H$ along the shear margin is lowest on the mesh adapted from $H$, the same meshes lead to highest errors in the vicinity of the grounding line and in the ice shelf, due to the grounding line migrating out of the fine-resolution region (see Fig. 8(a)). A more widely refined mesh generated from $\boldsymbol{u}$ is able to most accurately model the sharp velocity changes in the ice shelf, but does not leave enough resolution in the direct vicinity of the grounding line. Overall, the lowest error on both

$H$ and $\boldsymbol{u}$ is obtained on the mesh generated from $\boldsymbol{\tau}_b$, which contains enough resolution in the direct vicinity of the grounding line to be able to accurately represent it throughout the 10-year subinterval (see Fig. 8(d)). However, poorest solution accuracy is obtained along the grounding line flank along the shear margin and in the ice ridge, for all choices of adaptation sensor fields. This was also observed in Fig. 2, where the $0.5\,\mathrm{km}$ uniform resolution results most deviate from the reference $0.25\,\mathrm{km}$ resolution solutions in these areas.

Similar conclusions follow by investigating adapted meshes at other subintervals and error distribution at other times, which are then reflected in the time-averaged $\tilde{e}_H$ and $\tilde{e}_{\boldsymbol{u}}$ shown in Fig. 11. We observe that $\tilde{e}_H$ decays at approximately linear rate for all experiments, while the convergence of $\tilde{e}_{\boldsymbol{u}}$ tends towards a quadratic rate for adaptive mesh experiments. At low target

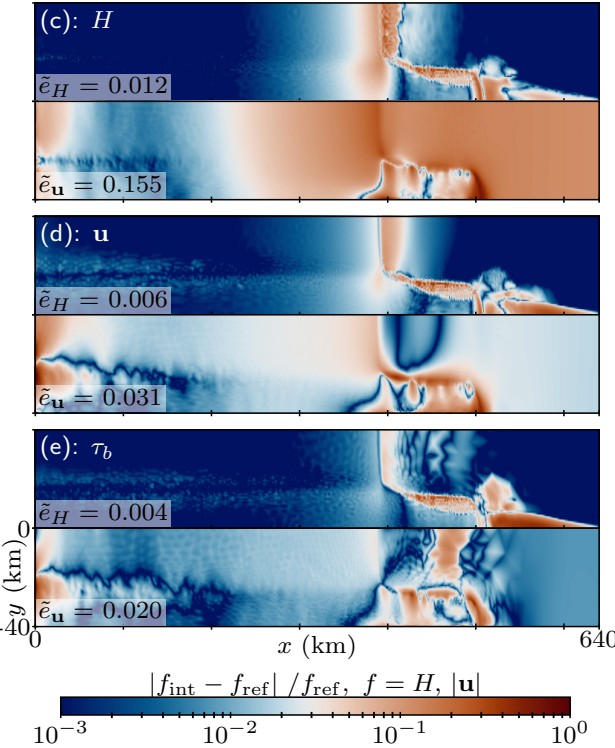

**Figure 10.** Spatial distribution of errors on $H$ and $u$ relative to 0.25 km uniform resolution solutions shown at $t = 100$ a for meshes $\mathcal{H}_9$ generated from the Hessian-based metrics of (c) $H$, (d) $u$, and (e) $\tau_b$ and $\mathcal{C} = 6400$.

complexities, mesh sequences generated from the basal stress $\tau_b$ yield the most accurate solutions since they most accurately represent the evolving grounding line. However, the increase in accuracy gained by further refining such meshes is effectively
capped, since the metric computed from $\tau_b$ does not prescribe refinement in the ice shelf. Nonetheless, we obtain the solution accuracy comparable to that of a uniform 0.5 km uniform mesh on a mesh sequence which has nearly 30 times fewer average number of vertices. On the other hand, errors computed on mesh sequences adapted from $H$, $u$, and their intersection are all similarly high for target complexities of 800 and 1600. Differences begin to be more pronounced for target complexity of 3200, when the metric from $u$ prescribes finer resolution in the vicinity of the grounding line compared to the metric from $H$. The
errors begin to be comparable to that of the uniform 0.5 km uniform mesh simulation at the target complexity of 6400, with 8 to 12 fewer number of vertices. Unlike the meshes adapted from $\tau_b$, the solution error for mesh sequences adapted from $H$ and $u$ keep decreasing for higher target complexities, as shown for the final target complexity of 12800.

### 4.4 CPU time efficiency

Even though we have been successful in outperforming uniform resolution meshes at a fraction of the total number of vertices,
this does not necessarily translate into a successful reduction of total computational time. Time-dependent mesh adaptation

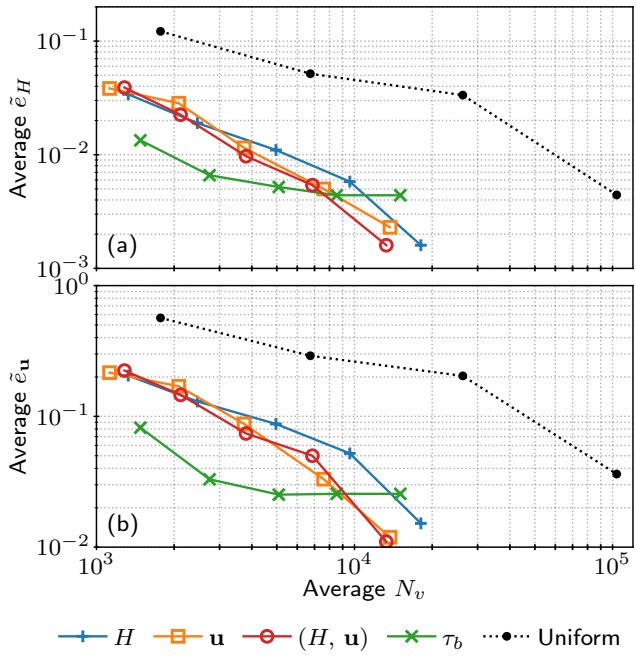

**Figure 11.** Accuracy of solutions computed on adapted meshes relative to 0.25 km uniform resolution solutions averaged over the simulation time interval of modelled (a) ice thickness $H$ and (b) ice velocity $\boldsymbol{u}$ for different sensor fields used in constructing the metric, and for increasing specified target complexity: $\mathcal{C}$, $2\mathcal{C}$, $4\mathcal{C}$, $8\mathcal{C}$, and $16\mathcal{C}$, where $\mathcal{C} = 800$. Solution accuracy of simulations involving uniformly refined meshes are shown for comparison.

routines presented in this paper imply additional costs related to metric computation, mesh generation, and function interpolations between meshes. None of these are present in single, fixed-mesh simulations. Moreover, the procedure is iterative, which involves repeatedly solving the PDE problem and adapting the mesh sequence until the mesh sequence has converged. Over all iterations of the hybrid mesh adaptation algorithm, the PDE problem was solved 4 times, the mesh sequence was adapted

4 times, and ice velocity and thickness fields were each transferred 80 times between subsequent meshes. The final decision behind which adaptation method (if any) to employ should therefore consider the total computation time necessary to achieve a desired solution accuracy. In Fig. 12 we therefore plot the average $\tilde{e}_H$ and $\tilde{e}_{\boldsymbol{u}}$ versus total CPU time for different choices of adaptation sensor field and for target complexities of 800, 3200, and 12800. Thanks to PETSc's and Firedrake's logging infrastructure, we are able to easily extract the time involved in each of the above-mentioned components, which we show in Fig.

13. As in the uniform resolution simulations, all simulations were run in serial for the purposes of CPU time measurements.

     As shown in Fig. 13, solving the PDE problem accounted for approximately only 50% of the computational time for $\mathcal{C} = 800$ and 75% for $\mathcal{C} = 12800$. This indicates that the efficiency of mesh adaptation methods is decreased for meshes with small number of vertices due to the relatively high costs of mesh adaptation and solution interpolation. However, as the solver or mesh complexity increase, the benefits of mesh adaptation become more pronounced. The cost of mesh adaptation is particularly high

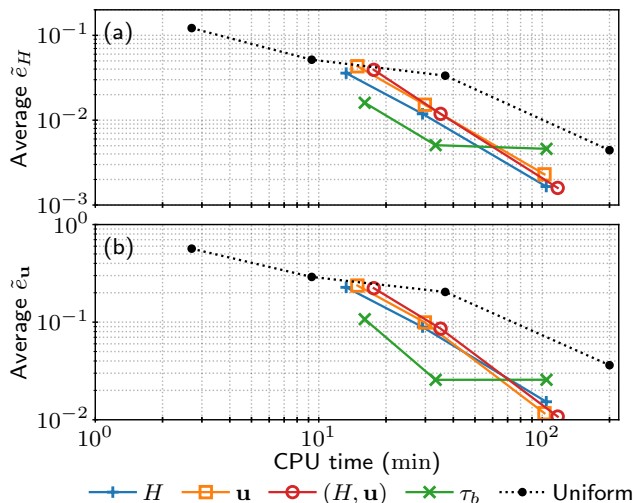

**Figure 12.** Performance and efficiency comparison of the hybrid mesh adaptation algorithm for different sensor fields and target complexities of 800, 3200 and 12800. Experiments were run on a single CPU and the algorithm terminated after 3 iterations. (a) Average $\tilde{e}_H$ and (b) $\tilde{e}_{\boldsymbol{u}}$ versus CPU time for different adaptation sensor fields: $H$, $\boldsymbol{u}$, intersection of $H$ and $\boldsymbol{u}$, and $\boldsymbol{\tau}_b$, compared to uniform mesh refinement.

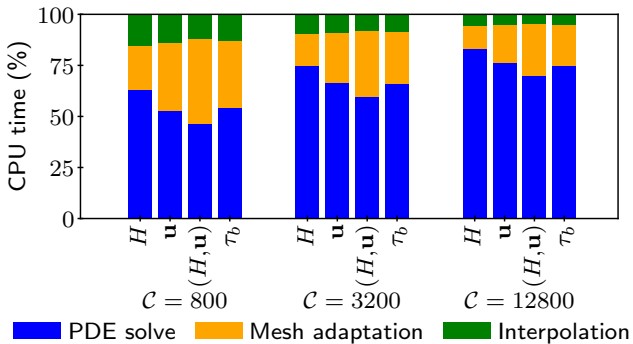

**Figure 13.** Distribution of CPU time across different computational tasks: PDE solve, mesh adaptation (including metric construction), and interpolation.

for vector-valued sensor fields since the Hessian metric of each component must be computed separately, and then intersected to form a single metric. Intersected metric of $H$ and $\boldsymbol{u}$ therefore implies the highest computational cost, since it involves 3 Hessian field computations (one for the scalar field $H$ and one for each component of the vector field $\boldsymbol{u}$) and 2 intersections for each metric (one for intersecting components of $\boldsymbol{u}$ and one for intersecting $H$ and combined components of $\boldsymbol{u}$). As a result, the superior performance of the intersected metric field $(H, \boldsymbol{u})$ observed in Fig. 11 is not reflected in the overall computational

efficiency shown in Fig. 13. Meanwhile, adapting the meshes based on $\boldsymbol{\tau}_b$ again clearly outperforms other sensor fields at low target complexities. Such adapted meshes achieve a solution accuracy comparable to the 0.5 km uniform resolution mesh with

a sixfold reduction of total computational time. All other sensor fields considered in this section achieve the same accuracy with approximately a threefold reduction of total computational time.

## 5 Conclusions

In this study we have demonstrated the effectiveness of anisotropic metric-based mesh adaptation for ice flow modelling. All libraries used here are Python-based and built on top of Firedrake, which, in combination, provide an intuitive and accessible approach to finite element ice flow modelling and mesh adaptation.

Using the setup of the Ice1 experiment of the Marine Ice Sheet Model Intercomparison Project (MISMIP+), we have shown that feature-based anisotropic mesh adaptation based on the Hessian of the solution-dependent field allows us to efficiently optimise the number of vertices in space and time. Our results confirm the importance of adequately modelling grounding line dynamics, and the necessity of fine mesh resolution in its vicinity. Adapting the mesh sequence based on the basal stress is therefore the most efficient option, since the finest mesh resolution is prescribed along the grounding line at a particular expense of a coarsened ice shelf, where the basal stress is zero. Meshes adapted from ice velocity fields are the most efficient at capturing the ice shelf dynamics, while ice thickness prescribed the most resolution along the shear margin and in the region of the ice shelf near the grounding line. All choices of the adaptation sensor fields were able to achieve accuracy in ice thickness within a few per mille, and the ice velocity magnitude within a few percent of the reference 0.25 km uniform resolution simulation results. Specifically, we show that adapted mesh sequences are able to reach the same solution accuracy as the uniform 0.5 km resolution mesh, but with 8–30 times fewer degrees of freedom. However, this translates into a 3–6 times lower computational cost, reflecting the cost of our iterative mesh adaptation procedure.

We emphasise the importance of the interpolation stage associated with each mesh adaptation. The bounded variant of the projection operator used in this paper introduces a potentially significant amount of numerical diffusion, which impairs the overall simulation accuracy. Future Animate development will focus on developing a post-processing routine in order to achieve a minimally-diffusive projection, as described in Farrell et al. (2009).

While results here have been obtained at a significantly lower computational cost using mesh adaptivity, this study reveals promising possibilities for further time reduction. As shown in section 4.4, the main contributor to the computational cost of the mesh adaptation methods employed here is their inherent iterative procedure. To that end, we have demonstrated one possible way of reducing the total number of iterations: by combining the classical mesh adaptation algorithm with the global fixed-point algorithm in order to obtain a more optimal initial mesh sequence. The presented hybrid algorithm required approximately 50% fewer iterations in order to reach mesh convergence than the global fixed-point algorithm alone, while still controlling spatial error and its temporal distribution. As noted in section 4.1, further computational gain may be obtained from a strategic time interval partition, which may reduce the total number of mesh adaptations and solution transfers. Based on these findings, our future work is focused on developing new mesh adaptation schemes inspired by the "on-the-fly" nature of the classical mesh adaptation algorithm, that will further reduce total simulation time. We expect that this will allow for even higher fidelity large- and global-scale glaciological modelling, where each iteration of the fixed-point mesh adaptation algorithm implies significant

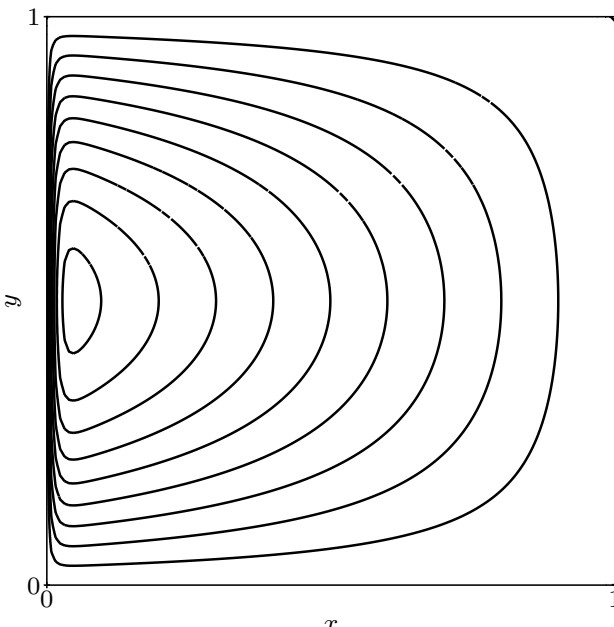

**Figure A1.** Contour plot of the manufactured solution $u(x,y) = (1 - e^{-x/\epsilon})(x-1)\sin(\pi y)$ to the Poisson problem (A1).

computational cost. However, more research is needed to investigate the effectiveness of these methods in larger-scale and less idealised models.

*Code and data availability.* All code and software used in this paper are free and publicly available. Animate and Goalie are available under the MIT license, with the exact versions used in this paper archived on Zenodo: https://doi.org/10.5281/zenodo.11537707 (Wallwork et al., 2024a) and https://doi.org/10.5281/zenodo.13353191 (Wallwork et al., 2024b), respectively. All results in this paper may be reproduced from

the source code hosted on Zenodo: https://zenodo.org/doi/10.5281/zenodo.14913405 (Dundovic, 2025). The repository contains scripts used for all presented computations and for generating all figures. Figures were created using Matplotlib (Hunter, 2007) with Scientific colour maps from Crameri (2023).

## Appendix A:  Poisson equation

Following a similar example as in Piggott et al. (2009a), we demonstrate the anisotropic metric-based mesh adaptation pre-
sented in this paper on a relatively simple example of a Poisson equation. We consider a unit square domain $\Omega = [0,1]^2$ and homogeneous Dirichlet boundary conditions. Using the method of manufactured solutions (Roache, 2001), we select the solution $u(x,y) = (1 - e^{-x/\epsilon})(x-1)\sin(\pi y)$, where $\epsilon = 0.01$, that is the solution to the following steady Poisson problem:

$$\nabla^2 u = \left( \left( 2/\epsilon - 1/\epsilon^2 \right) e^{-x/\epsilon} - \pi^2 (x-1) \left( 1 - e^{-x/\epsilon} \right) \right) \sin(\pi y). \tag{A1}$$

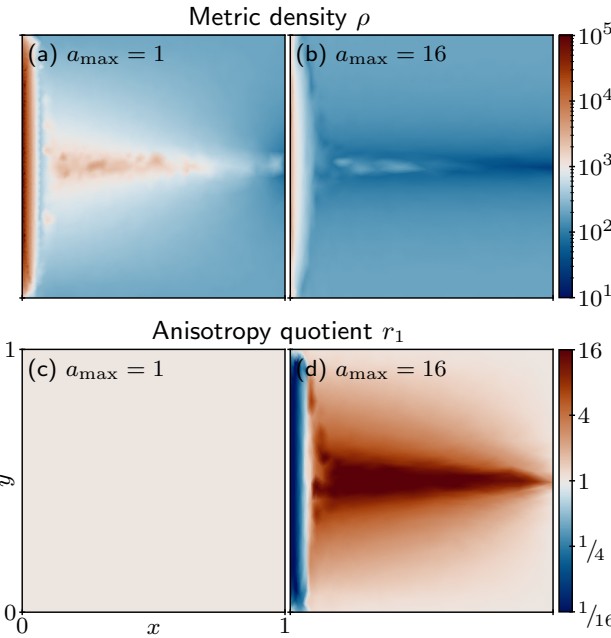

**Figure A2.** ((a) and (b)) Metric density and ((c) and (d)) anisotropy quotient $r_1$ of metrics defined from the Hessian of the solution of Eq. (A1), shown in Fig. A1, with maximum tolerated element anisotropy, $a_{max}$, of ((a) and (c)) 1 and ((b) and (d)) 16.

As shown in Fig. A1, the solution exhibits a sharp gradient in the $x$-direction near the $x = 0$ boundary since the term $1 - e^{-x/\epsilon}$ decays rapidly for $\epsilon = 0.01$.

The optimal adaptive mesh is found approximately, and by iteration. The process begins with a (coarse) uniform mesh on which we compute an approximation of the solution $u_h$. The mesh is then adapted and the solution $u_h$ is again computed on this new, adapted mesh. The process repeats until a set maximum number of iterations, or some other criterion, is reached. We expect a reasonable metric definition to lead to a more optimal mesh, which in turn leads to increased accuracy, particularly in the first few iterations when we transition from a uniform mesh to adaptive meshes.

The first step of a feature-based mesh adaptation process is the construction of a metric field from the Hessian of a chosen feature, i.e., sensor field, as described in sections 3-3.4. In the relatively simple example of a Poisson problem (A1), the only choice of a sensor field is the solution itself. We can approximately control several parameters during metric construction, such as the maximum tolerated metric anisotropy, $a_{max}$. By choosing the value of $a_{max}$ of 1 and 16 we obtain a pair of isotropic and anisotropic metric fields, respectively, whose components, metric density $\rho$ and anisotropy quotients $r_1$ and $r_2 = r_1^{-1}$, are shown in Fig. A2. Metric density is highest along the $x = 0$ boundary and the $y = 0.5$ midline in both metrics, but it is over an order of magnitude larger in the isotropic metric. As expected, anisotropy quotients of the isotropic metric are uniformly one, while the anisotropy quotients of the anisotropic metric vary spatially: $r_1$ ($r_2$) is 1/16 (16) along the $x = 0$ boundary and 16 (1/16) along the $y = 0.5$ midline.

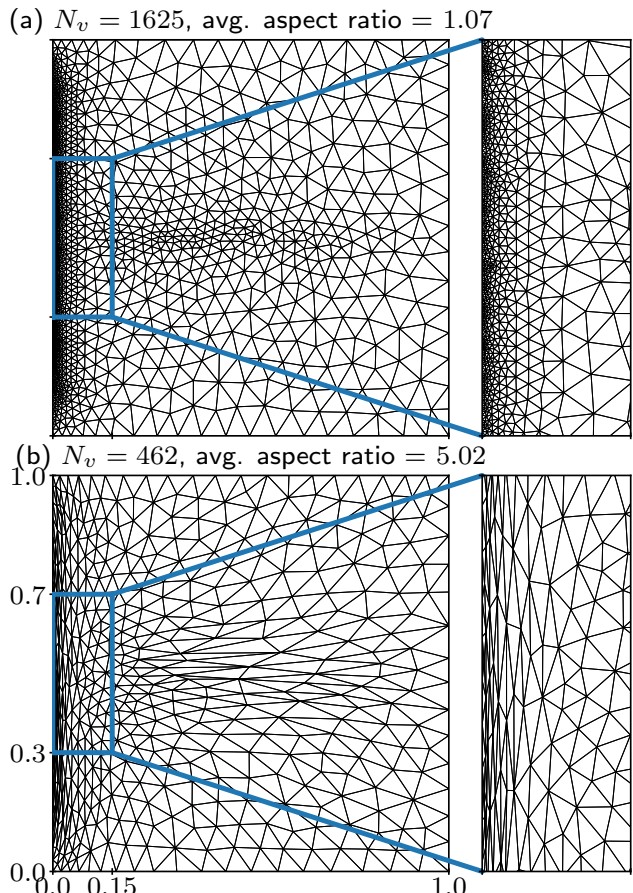

**Figure A3.** (a) Isotropic and (b) anisotropic meshes adapted based on the Hessian of the computed solution $u_h$ of Eq. (A1). A close-up of the domain region $[0, 0.15] \times [0.3, 0.7]$ shows fine resolution along the $x = 0$ boundary in both meshes. The isotropic mesh contains a much larger number of vertices $N_v = 1625$ compared to the anisotropic mesh with only $N_v = 462$, mainly due to the large number of isotropic elements along the $x = 0$ boundary.

As described in section 3.1, metric density $\rho$ controls the size of the mesh elements during mesh adaptation, while anisotropy quotients $r_1$ and $r_2 = r_1^{-1}$ control their shape. The influence of metric components is directly seen on meshes adapted from these metrics, shown in Fig. A3. In both cases the region near the $x = 0$ boundary is finely-resolved, but the isotropic mesh requires nearly four times as many vertices as the anisotropic mesh to do so. This is directly reflected in the respective metric densities, where the density of the isotropic metric was over a magnitude larger than that of the anisotropic metric. Since the variation along the $y$-direction is not as rapid as along the $x$-direction, elements of the anisotropic mesh are stretched along the $y$-direction, thus avoiding unnecessary over-refinement along the $x = 0$ boundary. We further observe refinement along the $y = 0.5$ midline, with the metric density and resulting number of elements again being higher in the isotropic case.

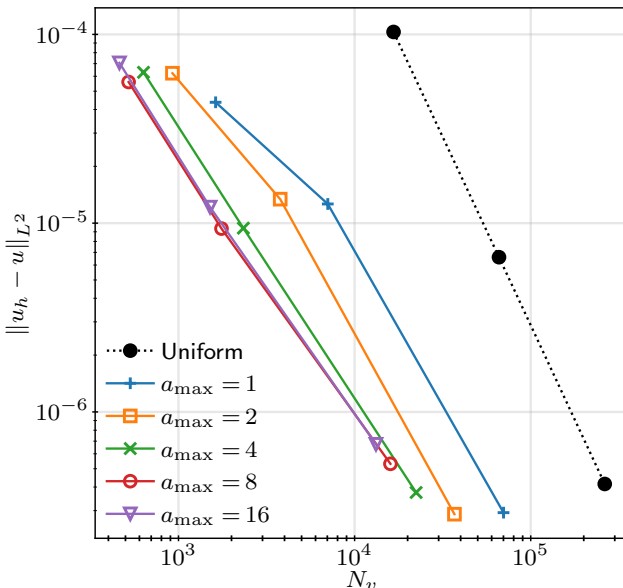

**Figure A4.** Error in the numerical solution $u_h$ to the Poisson problem given in Eq. (A1) against the number of vertices $N_v$ of the mesh on which it was computed. Maximum tolerated metric anisotropy values $a_{\max} \in \{1, 2, 4, 8, 16\}$ were prescribed during metric construction.

Since the analytical solution is available, we can easily compute the accuracy of the computed solution $u_h$ by comparing it to the analytical solution $u$ in $L^2$ norm. The error of the solutions computed on the isotropic and anisotropic meshes in Fig. A3 is roughly the same ($4.37 \times 10^{-5}$ and $7.07 \times 10^{-5}$, respectively), even though the anisotropic mesh has roughly four times fewer vertices. We can conclude that there was indeed no significant benefit from finely-resolving the region near the $x = 0$ boundary along the $y$-direction.

We repeat the process again by considering intermediate values of $a_{\max}$ and by varying the total number of vertices of generated meshes. For comparison, we also solve the problem on uniform meshes. For each solution we compute the error and plot it in Fig. A4 against the number of vertices of the mesh on which it was computed. We observe that adaptive simulations achieved the same error as uniform simulations with over an order of magnitude fewer number of mesh vertices. In particular, there is a clear reduction of error for the same number of vertices when considering increasingly anisotropic meshes. However, we do not observe a clear improvement of increasing $a_{\max}$ from 8 to 16. Had the solution $u$ had an even steeper gradient, we would expect to see benefit from further anisotropy.

Uniform refinement achieves the expected theoretical second-order convergence rate ($\mathcal{O}(N_v^{-2})$), while adaptive refinement achieves slower convergence rates, between $\mathcal{O}(N_v^{-1.35})$ and $\mathcal{O}(N_v^{-1.48})$, although it should be noted that there is no longer a simple relationship between $N_v$ and (minimum or average) element size for the adapted meshes. The errors on uniform and adaptive (isotropic and anisotropic) meshes would converge for high enough $N_v$, when even the uniform mesh is fine enough near the $x = 0$ boundary, but as well as other regions where such fine resolution is unnecessary. Anisotropic mesh

adaptation prevents such unnecessary over-refinement by distributing the available resolution where it is necessary, according to the prescribed metric. Such an approach ensures that available computing resources are efficiently utilised.

*Author contributions.* All authors jointly developed the concept of the study. DD designed the novel hybrid mesh adaptation algorithm, conducted all analyses, ensured reproducibility of results, and created all figures. JW is the original author and main developer of the Animate and Goalie software, to which DD contributed significantly by extending existing features and implementing new ones, with input
from JW and SK. MP and RH supervised the work. RH was the Principal Investigator of the study, contributed to the glaciological context of the work, and together with DD secured funding for the research. DD prepared the initial draft, with contributions by FGC in writing the introduction section. All authors commented on and contributed to improving the clarity of the manuscript.

*Competing interests.* The authors declare that they have no conflict of interest.

*Acknowledgements.* DD and RH were supported by the Norwegian Research Council (Norges Forskningsråd) Project 324131. DD was
further supported by the Norwegian Artificial Intelligence Research Consortium (NORA.ai) and The Alan Turing Institute's Enrichment Scheme. RH was further supported by ERC-2022-ADG grant 101096057. The computations presented in this paper were performed on the Norwegian Research and Education Cloud using resources provided by the University of Bergen and the University of Oslo.

The authors thank Daniel Shapero for his input and help on setting up simulations in icepack, and Eleda Johnson for sharing her mesh adaptation experience and for reviewing new contributions to the software.

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
