# Peer review of "Anisotropic metric-based mesh adaptation for ice flow modelling in Firedrake"

_EGUsphere, 2024_

## Author Response (AR1)

**Anisotropic metric-based mesh adaptation for ice flow modelling in Firedrake**

This document provides a point-by-point response to the comments raised by Reviewer #1.

*Response to Reviewer #1*

*We thank the reviewer for the detailed review! Original comments are written in black, while our responses are provided in blue.*

Summary: This manuscript describes, implements, and tests an adaptive mesh refinement (AMR) scheme for the 2D time-dependent shallow shelf approximation (SSA) models of marine ice sheets. The scheme is based on metric-based mesh adaptation, using fields derived a posteriori from ice thickness, ice flow speed, the magnitude of basal shear stress, or intersections thereof, as the local refinement criterion. The essential idea is that the adapted mesh comes from solving an optimization problem for a Riemannian metric field, constrained by a fixed number of mesh nodes, using Hessians (2nd derivatives) of the current numerical solution. The scheme has an open-source Firedrake-based Python implementation, with mesh construction from the computed metric ("mesh adaptation") via the library Mmg2d. The AMR scheme is tested on the MISMIP+ marine ice sheet idealized experiment. With the caveat that no MISMIP+ exact solution is known, clear reduction in mesh complexity is observed, relative to uniform mesh refinement, for a given numerical error level. Modest speedup, in a time-to-error sense, occurs in the right circumstances, but the large cost of computing solution Hessians is noted.

Recommendation: The ideas here, and I believe the implementation (though I have not tested it), represent valuable work. Concretely, an open source, Firedrake-based toolchain for adding principled (metric and anisotropic) AMR to marine ice sheet simulations is a *good thing*. The reported performance results suggest a meaningful speed-up over a uniform mesh, though apparently less than an order of magnitude. (In this regard, speedup must at least compensate for the increased complexity.) This work should be published in a well-written and self-contained form. However, while the authors intend that the current manuscript is "self-contained", in fact it is fundamentally lacking in this regard. Almost all readers will have difficulty in following the technical details; only those with prior metric-based adaptation experience could be comfortable. Serious readers will split into those who willing use the authors' implementation, trusting that things work in some reasonable manner, and a (smaller?) group who must reconstruct for themselves, from the literature and the source code, many aspects of how things actually work, either in high-level outline or in detail. I care about the latter group; please see the suggestions following. I would suggest that the paper be returned for major revisions.

Suggested presentation, so as to be self-contained: The content in sections 2.1–2.4 is reasonably self-contained, but it would be more so if it explicitly described how things work for the familiar Poisson equation, presumably with the solution (absolute value of the solution?) as the "sensor field". A picture of how a mesh is actually generated from a metric is currently missing as far as I can tell, and it could be easily added to this Poisson case; it is the main point. The fact that actual mesh adaptation occurs inside Mmg2d should be made clearer to the casual reader. A clear visual image of metric and mesh anisotropy is also missing, and this could be added. Now a new section 3 could be created containing the marine ice sheet equations (which should be stated as a coherent, essentially complete block), the new time-dependent ideas, and a much clearer discussion of the nontrivial sensor field choice for marine ice sheets. These new things are the novel ones, relative to prior anisotropic metric-based AMR, as far as I can tell. Such important new concepts should not be confusingly scattered in the way they currently are. It must be clearly indicated to the glaciologically-inclined reader, presumably in the new section 3, and in any case well *before* the end of the results section, that vector-valued SSA-based marine ice sheet simulation requires a nontrivial physics choice regarding AMR, and that this choice is not a settled matter. That is, one needs to *choose* between surface speed, ice thickness, basal stress, intersected speed/thickness, and etc as the sensor field. (The current manuscript assumes glaciological readers will happily read all the way to page 20 and 3.4.2 before having any clear statement of the physics basis on which the mesh is refined! This is crazy.) The same choice should be clearly sketched in the Introduction, and perhaps even the abstract. Then, in section 4 on results, the particular MISMIP+ experimental configuration could be described, and finally the actual results given. Finally, many of the Figures are

elaborate in the modern Nature/Science style, and thus no reader understands them; Figures 5,9,10 are this way and I suggest they be broken into discrete digestable form. The spaghetti needs to be unwound.

*We take the liberty of summarising the above comments under the following points:*

1. *Demonstrate the mesh adaptation process on a simpler example of a Poisson equation.*

2. *Introduce glaciology-related discussions earlier in the manuscript, especially those related to the choice of sensor fields.*

3. *Add (more) figures demonstrating metric components and the resulting adapted mesh.*

4. *"The fact that actual mesh adaptation occurs inside Mmg2d should be made clearer to the casual reader."*

5. *"Finally, many of the Figures are elaborate in the modern Nature/Science style, and thus no reader understands them; Figures 5,9,10 are this way and I suggest they be broken into discrete digestable form."*

*Point 1: While we believe that our work is relatively easy to reconstruct, both from our published code and open source software that we used, we acknowledge its complexity and agree that a familiar example of a Poisson equation would be beneficial to those new to the idea of mesh adaptation. However, adding a new section demonstrating this simple setup, as well as adding multiple figures describing the mesh adaptation process on the Poisson equation, would push the glaciological discussions even further back in the manuscript, which is contradictory to point 2 above. We have therefore added the Poisson based example as a new appendix.*

*Point 2: We moved subsections 3.1 and 3.2, which describe the MISMIP+ experiment setup and resolution sensitivity study on uniform meshes, to a separate section that follows the introduction. This allowed us to, early-on, motivate glaciological discussion and demonstrate why one should care about mesh adaptation. We also introduce some fundamental ideas of metric adaptation, including the notion of sensor fields, which we immediately connect with the presented MISMIP+ experiment.*

*Point 3:*

- *While a figure demonstrating how a mesh is generated from a metric is in fact present (Figure 1), we have expanded this figure to now show three different metrics as well as the meshes adapted from each of those metrics.*

- *In the above-described figure we further zoom-in on different regions of the mesh to better demonstrate what the anisotropy quotients represent.*

- *Following the same format as the figure described above, we have added a new figure demonstrating a metric constructed by intersecting two metrics. A corresponding adapted mesh is also shown.*

*Point 4: This point has been incorporated within the abstract.*

*Point 5: Each one of these three figures has been split into two figures.*

1. line 2,3 (in abstract): Avoiding the boring sales pitch about "comprehensive overview of state-of-the-art" stuff. Perhaps "In this paper we propose feature-based anisotropic mesh adaptation ..."

   *We have modified the sentence as suggested. One of our aims was to formally introduce anisotropic metric-based mesh adaptation to the glaciological community, to which we dedicated a significant portion of the paper (sections 2.1-2.6). We wanted to reflect this in the abstract.*

2. line 6: Remove "more optimal"; it does not mean anything. In the same line, "50% fewer" than what? Say briefly.

   *Removed "more optimal" and modified the sentence to "... that reaches mesh convergence approximately twice as fast compared with the existing global fixed-point algorithm."*

3. line 11: Almost no readers (me included) know what a "sensor field" is. This needs a rewrite in the abstract. I suggest avoiding "sensor field" here (but paraphrase it), and then carefully define it later, e.g. at line 181, which seems to be the next use of the phrase.

   *We now completely avoid the term "sensor field" in the abstract. Instead, following your comment 77, we explicitly state that we consider basal stress, ice thickness, and ice velocity in metric construction.*

4. line 11: What does "Due primarily to the iterative nature" mean here? The reader is unlikely to believe that just because it is iterative it is therefore faster or better. Perhaps this last sentence should be something like: "In practice the adapted mesh technique translates into a ..."

   *We have modified the sentence to now read: "Since the fixed-point algorithms require that the problem is solved multiple times, the reported reduction in number of vertices ultimately translates into a 3–6 times lower overall computational cost compared to uniform mesh simulations."*
   *In this sentence we in fact wanted to emphasise the slowness of this iterative mesh adaptation process, rather than the opposite. By emphasising the iterative nature of the procedure, we wanted to portray that applying the hybrid algorithm has made it half as expensive (i.e., twice as fast) due to having halved the number of iterations needed to reach mesh convergence. In the conclusion we expand on this further and say that our future work is focused on even further reducing the number of iterations needed to reach mesh convergence.*

5. lines 14-81: I think the Introduction is quite well-written, and it accords with my understanding of the development of models suitable for marine ice sheets.

   *Thank you.*

6. line 22: For the reader to understand the meaning here I suggest: "... the majority have focussed on reducing computational costs by numerical and meshing techniques, while maintaining the model physics."

   *Thank you; we have modified the sentence exactly as proposed.*

7. line 27: Recent significant progress has been made on this contact problem. See: de Diego, G. G., Farrell, P. E., & Hewitt, I. J. (2022). Numerical approximation of viscous contact problems applied to glacial sliding. Journal of Fluid Mechanics, 938, A21.

   *We now cite this as well.*

8. line 29: It is not clear what "Alternative" means here. This lead has the same meaning?: "Mesh adaptation methods select finer ..."

   *Changed to "On the other hand, mesh adaptation methods select finer..."*

9. line 65: Suggest "allows to equidistribute" –> "equidistributes".

   *Done.*

10. line 72: "velocity norm" –> "speed"

    *Done.*

11. lines 73–75: I had to reread the last two sentence of this paragraph several times. I suggest this 3 sentence replacement: "These methods generate only a single mesh that is used throughout the simulation. However, they do not consider the nonlinear and time-dependent coupling between the solution and the underlying mesh. This coupling suggests an iterative mesh adaptation procedure, which we will implement."

    *Thank you; we agree this is clearer and have modified the sentences as suggested.*

12. line 76: As noted earlier, my main concern about the current manuscript is that it is *not* a sufficiently self-contained description, given the reader audience, including myself.

    *We respond to this in detail under your Recommendation paragraphs (Points 1 and 3).*

13. line 79: "but implement" –> "and we implement"

    *Done.*

14. line 83: Presumably $\Omega$ has a polygonal boundary.

    *Indeed. We have now specified that the domain is polygonal.*

15. line 88-89: It is true that there is not a "monotonic" relationship, but that is hardly the point. AMR does not guarantee monotonicity here either, as far as I know. The point is that uniform refinement can lead to only small, thus expensive, reductions in the error.

    *We agree that talking about the relation between $N_v$ and error not being monotonic can be confusing. We have now dropped this.*

16. equation (1), page 4: The displayed equation should be taken more seriously. It should say something like "find $H_{opt}$ solving min $E(H)$ over $H$ with $N_v$ vertices". Then the sentence after can be shortened.

    *Done.*

17. lines 96-97: Suggest: "In anisotropic mesh adaptation the optimal mesh is only found approximately, and by iteration."

    *Thank you; we have modified the sentence as suggested.*

18. line 102: "a way" –> "a local way"

    *Done.*

19. lines 103-104: Actually put the definition after the sentence which promises the definition: "A Riemannian metric is a smooth function yielding a positive-definite 2x2 matrix M(x) at each x in the domain."

    *Thank you; that indeed improves the flow of the text. We have combined the sentence from line 109 with the sentence in this line. It now reads: "The notion of measuring distances arises from the definition of a Riemannian metric $\mathbf{M}(\vec{x})$, which is identified with a positive-definite matrix in $\mathbb{R}^{d\times d}$ at each point $\vec{x}$ of the d-dimensional domain."*

20. somewhere around equation (2): State that $\lambda_1 \geq \lambda_2$ somewhere, as I am pretty sure you are using that.

    *We added this to the sentence where the eigenvalues were introduced (the sentence leading to Eq. (2) in the original manuscript).*

21. line 115: "a metric tensor M is a continuous analogue of a single mesh element" makes no literal sense. Probably: "a metric tensor field M(x) is the continous analog of the changing shape [dimensions?] of elements as one moves through the mesh." Yes? Otherwise "continuous" is not modifying anything?

    *Thank you; this was indeed confusing. We have simplified the sentence to now read: "In the continuous metric framework, the metric $\mathbf{M}$ prescribes the optimal size of the mesh element (through density $\rho$), its optimal shape (through quotients $r_1$, $r_2$), and its orientation (through eigenvectors in $\mathbf{R}$)."*

22. lines 119-120: "fully" makes no sense here except as vacuous advertising. How about "... in R), thus the metric-based approach allows anisotropy in the resulting mesh."

    *We have modified the sentence to now read: "Thus, the metric-based approach introduces anisotropy in the resulting mesh."*

23. caption Figure 1: "is 16 times" –> "is in fact 16 times"

    *In response to this comment and the comment below, we have modified this sentence in the caption to now read: "The zoomed-in subfigures maintain equal scaling of the x and y axes, while the domain is laterally shortened for legibility in other subfigures."*

24. caption Figure 1: ", which is not reflected in the figure for readability" –> "; the domain is laterally shortened for legibility."

   *Done (see comment above).*

25. line 121: "metric space" actually means something, which is not here. How about "... the Riemannian space $M = (\Omega, M(x))$ models ..." Surely one can call M(x) a metric tensor field, or just a metric field, and not conflate the ideas of "metric space" and "Riemannian manifold"?

   *Changed "Riemannian metric space $\mathcal{M} = \{\mathbf{M}(\mathbf{x})\}_{\mathbf{x} \in \Omega}$" to "metric tensor field $\mathcal{M} : \mathbf{x} \in \Omega \mapsto \mathbf{M}(\mathbf{x})$". We think this notation for $\mathcal{M}$ is more appropriate than our original notation, since it is not just a set of loose tensors labelled with arbitrary positions.*

26. line 126: "is of" –> "has"?

   *Done.*

27. line 127: Since I actually know what a Riemannian manifold is, and what a metric space is, I am again irritated that you could say "Riemanninan space" or "Riemannian manifold" here without harm, but instead you have a jumble. The metric space structure of a Riemannian space/manifold comes from minimizing the integral of the Riemannian metric over curves, right? But you don't need that metric space structure, I am quite sure. In conclusion, avoid "metric space" unless that is really what you mean.

   following up: I see that the fundamental confusion is already present in Alauzet 2010. That author simply decides to measure distances between points in a "Riemannian metric space" in a way which is different from the metric space structure of a Riemannian space. He simply decides that you integrate over Euclidean lines. Indeed he does not need the correct metric space, and just overwrites it with his stuff using the same words. It is sad and destructive that that author has now generated a branch of mathematics where words mean different things from the rest of math. (You will never be able to do metric-based adaptation on a Riemannian manifold without undoing his language choice.) Why did he call his thing "Riemannian" when he announces that it is not? I would be cool with "Alauzet metric space".

   *Thanks for pointing this out; we agree. We changed all occurrences of "Riemannian metric space" and "Euclidean metric space" to "Riemannian space" and "Euclidean space", respectively.*

28. line 135: Just strike "which appear to be ...". The shortening should be clearly stated in the Figure caption, e.g. as suggested above. Then do not refight that battle; it is confusing to do so.

   *Done.*

29. line 148: $||u - u_h||$ is *not* the "local interpolation error". Correct this; I think you are merely talking about $||u - \Pi_h u||$?

   *Thank you; this was indeed a typo. Fixed.*

30. end of line 153: "interpolant", right? (I see Alauzet already makes this English mistake.)

   *Modified to "interpolant" here and in line 143.*

31. equation (6): How about

   "find $M_{L^p}$ which minimizes $E_{L^p}(M) = \int_{\Omega} ...$"

   That is, the input variable to the objective is (caligraphic) $M$. Putting the $L^p$ subscript on "$M_{L^p}$" then labels this as the minimizer of $E_{L^p}$.

   *Thank you; modified as suggested.*

32. line 164: What does "smooth" mean? Drop that word? This is an $L^2$ projection into a continuous FE space, right?

   *Thanks for pointing this out. You're right, it is indeed sufficient to just drop "smooth". We made this change now.*

33. around lines 155-164: I think it should be said that you don't actually have access to $u$ and you are applying the idea to $u_h$ a posteriori.

*Thanks; we added this two sentences after Eq. (6) in the revised manuscript.*

34. line 166: At this point the reader has no idea what you will later mean by "one partition of the simulation time interval". Just refer to a "time interval" here, and don't burden the reader with your later optimizations.

*We have modified this sentence to now read:"... each mesh is assigned to a different part of the simulation time interval." We believe that using more plain language and avoiding the term "partition" makes this sentence less burdensome now. However, it was still important to mention that it is a distinct part of the overall time interval since Eq. (9) talks about computing the metric $\mathbf{M}^i$ for each subinterval $i$.*

35. line 176: "may necessitate" $->$ "needs"

*Done.*

36. line 181: At some point you need to define "sensor field". As done here it is either too late or too awkwardly said. Maybe connect to equation (6) at the beginning of this paragraph; say that in the scalar Poisson context one uses u for adaptation, and one calls this use the sensor field. Now say that there are choices in the time-dependent coupled context.

*We respond to this in detail under your Recommendation paragraphs (Point 2).*

37. line 184: "mean" $->$ "method"

*Done.*

38. equation (10): Here I think this is only well defined if $\lambda_1 \geq \lambda_2$, right? See comment about equation (2).

*We added $\lambda_1 \geq \lambda_2$ as per your previous comment (comment 20).*

39. line 190: "eigenvalues" $->$ "eigenvalue estimates"? I don't see how they can be eigenvalues unless the two metrics are assumed to be simultaneously diagonalizable by the $\{e_i\}$?

*Thanks for pointing this out. Instead of saying "eigenvalue estimates", which might be a bit vague again, we simply end the sentence with "... and $\lambda_i^j = \mathbf{e}_i^\mathsf{T} \mathbf{M}_j \mathbf{e}_i$." That is, we do not give a name to $\lambda_i^j$.*

40. lines 191, 192: strike "popular" and "commonly"

*Done.*

41. Figure 2: Note this is a detailed figure about a fiddly detail. It should go later, as part of putting novel ideas later. A couple of early figures emphasizing the main ideas of metric-based anisotropic mesh adaptation should be substituted.

*We agree that this is fiddly. The other reviewer commented that they did not find the figure helpful. We have therefore removed this figure and presented the algorithms as pseudocode, as suggested by both reviewers.*

*Early figures emphasising the main ideas of mesh adaptation were added, as we responded in detail under your Recommendation paragraphs (Point 3).*

42. section 2.4: This immediate descent into details of postprocessing misses the main message: from the a posteriori computed metric estimate $M(x)$, or $M^i(x)$ in the time-dependent case, you are asking Mmg2d to generate mesh(es). *Then* talk post processing details, though probably put later.

*The postprocessing steps described in this section modify the metric *before* the adapted mesh is generated by Mmg. That is to say, it is not the newly-generated mesh that is post-processed, as we understood from your comment. We believe that this is clear from the section name ("Metric*

*postprocessing"*) *and the first sentence in this section: "... post-process the metric field before the adapted mesh is generated." We have therefore decided not to make any changes to the text.*

43. line 238: Speaking as a reader, I would want the "several shortcomings" to be explained here. Evidently I am too stupid to remember the brilliant exposition from the Introduction.

    *We have paraphrased the three shortcomings mentioned in the introduction (lines 41–43 of the unrevised manuscript) and included them here.*

44. line 245: Interleaving the explanation of the algorithm and advertising its benefits is not helpful. Do the former first.

    *Done. Detailed response given under comment 47 below.*

45. line 247: Probably clearer without "before they are adapted".

    *Agreed; thanks. Removed this.*

46. line 248: Again, advertising is mixed into algorithm description. Not helpful.

    *Done. Detailed response given under comment 47 below.*

47. line 252: On first pass I definitely did not understand the meaning of "The fixed-point algorithm iterates ..." *here*. (I wrote: """Is this recomputing solutions u along the way? Are you returning to the start of the time partitions to revise the metric, or is the metric getting updated only in a feed-forward manner. Iteration is an ambiguous word, but your use of "fixed-point" suggests things are not feed-forward only. Does the "x" symbol in Figure 2 mean "compute a new metric but don't change the mesh"?""")

    *Thanks for pointing out that these paragraphs required clarifying. We modified the sentence to now read: "The algorithm terminates..."*
    *Furthermore, we believe that presenting the two algorithms as pseudocode helps clarify the thoughts you had written down. We also tidied up the text, which included some duplicated statements, as well as some tangential comments (i.e., the mention of "complexity ramping" in the unrevised manuscript). The section is therefore also significantly shorter, which should aid reading.*
    *And to respond to the final question: the "x" symbol in the Figure indeed meant what you wrote, but we decided to remove this Figure altogether (see our response to comment 41 above).*

48. section 2.5 generally: The first 4 paragraphs are quite good. Starting in the 5th paragraph the job is to describe unambiguously an algorithm. A pseudocode may be better than the paragraph-with-interleaved-commentary-and-advertising current form. Figures showing the action on coarse meshes might help.

    *We now present the two algorithms as pseudocode as suggested by both reviewers. We have also modified the text for clarity, as described in the four points above (44-47).*

49. generally: These time-dependent mesh adaptation schemes seem to be awkward, and early-development. Do you really recommend them for production simulations?

    *Existing time-dependent mesh adaptation algorithms certainly leave much to be desired. This is what motivated us to introduce the new "hybrid" algorithm, which we found to outperform existing algorithms described in this section. We believe that we have presented an honest assessment of available algorithms (including our own hybrid algorithm), but it is up to the user to decide which approach to adopt (if any); especially since this decision is problem-dependent. In the Conclusions section we further comment on this and state that we are focused on developing mesh adaptation algorithms that are more suitable for time-dependent problems.*

50. section 2.6: This part makes sense to me, but how understandable is it if you are not already comfortable with supermesh ideas, interpolation, and Galerkin projection?

    *We now refer readers to Farrell, 2010 for an overview of these methods, which also discusses their implementation through supermeshing.*

51. line 290: The avoidance of negative values via $P^1$ makes sense, but I suppose this begs the question of whether one should, instead of h-refinement and the costs of AMR, instead p-refine in a way which avoids the out-of-bounds problem. See recent work: Kirby & Shapero 2023. High-order bounds-satisfying approximation of partial differential equations via finite element variational inequalities arXiv:2311.05880

    *In response to comment 6 of the other reviewer, we have briefly expanded on some of the pros and cons of p- and h-refinement in section 3.2 of the revised manuscript. We have also cited the work you recommended.*

52. line 305: Give some citation for "PETSc's Riemannian metric functionality"?

    *We now cite Wallwork et al., 2022.*

53. line 308: As a nonexpert I don't know what "mesh optimization" means here. Which of the many ideas relating to building better meshes does it refer to?

    *We changed "Mesh optimisation" to "The process of locally adapting meshes with respect to metric fields...". This is further clarified in the following sentence (left unchanged).*

54. sections 2.6, 2.7: Easy to read.

    *Thank you.*

55. line 340: "the equations of glacier flow" should actually be stated. (The "the" is ridiculous here; it is a field with lots of models.)

    *We now briefly present the equations and refer the reader to Greve and Blatter, 2009 for a detailed discussion and derivation of the equations. Also removed "the".*

56. line 342: Indeed this alternation is "as with most current ... models". The scheme is called the explicit Euler approximation to the coupled model, that is, the scheme is explicit and first-order! As with current ice sheet models, attaching the "prognostic" and "diagnostic" language, a leftover of pre-numeric weather forecasting, is not helpful. The current authors are not to blame here, but they don't have to climb into the dumpster too.

    *We indeed adopted the terminology because it is so widely present throughout glaciological models, including icepack. Thank you for providing wider context of the terminology. We modified the text to avoid using "prognostic" and "diagnostic" terms throughout this section.*

57. line 353: Arg. Is "P" the pressure or the effective pressure? Effective pressure is the *difference* between the ice pressure (overburden pressure) and the pressure of subglacial water. At the bottom of floating ice this value is zero; assuming h is ice thickness and $d_W$ is the depth below the water line of the bottom of the ice, then $\rho_i g h = \rho_W g d_W$ because of flotation. Please don't use "P" for the effective pressure. Fix all this.

    *We have renamed effective pressure from $P$ to $N$ and clarified the text.*

58. line 356: Please do not assert that the scheme in IcePack is unconditionally stable. It is not. Importantly, Shapero (2021) does not assert that it is. They use a scheme which is unconditionally stable *for the linear advection equation*. Then they say: "We note that, while the stability properties of different schemes for the linear advection equation guided our choice of method, the coupled system for both thickness and velocity is not linear and not hyperbolic." The scheme in IcePack is implicit for the linear advection equation but *not* implicit for the coupled system. In this context it completely suffices to say that "Finally, IcePack implements a second-order Lax-Wendroff scheme for transport (Shapero, 2021)."

    *Thank you for the detailed explanation; we have modified the sentence as suggested except that we dropped "second-order". We think that this is not an important detail for our work and that it may cause confusion, as it did with the other reviewer. We instead clarify that the Euler approximation to the coupled model is first-order, as noted in your earlier comment 56.*

59. Figure 4: If this result is worth showing for uniform refinement, then also show it for a preferred AMR strategy? Presumably, with fewer dofs than the 0.25km uniform?

*The preferred AMR strategy involves $\vec{\tau}_b$ as the sensor field, so we use it to show the requested result in Fig. R1 below. However, after careful consideration, we have decided not to include it in the revised manuscript.*

*The motivation behind Fig. 4 (of the unrevised manuscript) is very different to what we present in the section on adaptive results. Fig. 4 was produces as part of the resolution sensitivity study, where, in the absence of an analytical solution, the aim was to quantify the differences between hierarchically refined meshes. We concluded that results have sufficiently converged for the resolution of $250\,\mathrm{m}$, which we then use to define a "reference" solution. In other words, we would immediately compare our uniform resolution results to an analytical solution had it been available, or a reference solution had a notion of it existed at that time. Defining the reference solution in turn allows us to define diagnostic fields and quantities such as $\tilde{e}_H$ and $\tilde{e}_{\vec{u}}$, which are more instructive and allow us to make more detailed observations. We believe that we have done so effectively. All takeaway messages of Fig. R1 had already been reflected in other figures and emphasised throughout the text.*

*Therefore, we do not think that this figure provides any further insight into our results. Furthermore, the section on adaptive results now contains 6 figures (after the splitting of figures recommended in comment 64), so we were hesitant to add another large figure, especially one whose purpose is not immediately clear.*

[Figure]

Figure R1: Comparison of uniform mesh simulations with adaptive simulations, where meshes were adapted based on $\vec{\tau}_b$ for target complexities of 800, 1600, and 3200: (a) volume above flotation ($V_f$) over time, (b) grounding line position along the midchannel line ($x_{gl}(y=0)$) over time, and (c) grounding line position in the domain at $t = 100\,\mathrm{a}$ and (d) $t = 200\,\mathrm{a}$.

60. line 363-364: Me too. You don't need to include this detail particularly, but the cp linesearch in PETSc never seems to work for me, and I fall back to backtracking.

*We found that cp linesearch was successful in all our uniform-mesh experiments, but sporadically unsuccessful on adapted meshes, as stated in this sentence. We therefore initially included it for the sake of completeness and transparency in assessing the application of mesh adaptivity. However, we have now removed it for the sake of conciseness, as recommended, and simply state: "Backtracking line search is used to solve non-linear systems."*

61. line 390: "emulate a more general problem". Huh? This sentence probably means "In the mesh adaptation experiments to follow, neither exact nor high-fidelity solutions are assumed to be available." On the other hand, a couple of sentences later you treat the 0.25km uniform result as high-fidelity. Rewrite this paragraph?

    *We have rewritten this paragraph to avoid confusion related to our intention to emulate a more general problem, which we now do not mention at all.*

62. line 396: I don't know what "effectively ... unbounded" means, just after you give bounds. Frankly, I have a sense that something is numerically unstable etc., and you are special pleading.

    *We agree that this was written ambiguously and have now modified the sentence to simply read: "We [...] do not prescribe bounds on element edge lengths and element anisotropy." The intent behind the original description was to say that the minimum and maximum element length bounds of 1 m and 50 km have not been reached in any of our experiments. In other words, none of our results generated adapted meshes with element lengths smaller than 1 m or larger than 50 km. That is to say, the same results are obtained with the default "unbounded" minimum and maximum element lengths of $10^{-30}$m and $10^{30}$m, respectively (i.e., they are "unbounded" in practice since these minimum and maximum lengths are never reached). It was not intended to hide any numerical instabilities, as you were concerned.*

63. section 3.3 generally: I think you can write up your experiment choices more declaratively and less defensively. To first approximation, just say what you did.

    *The frequent inclusion of explanations on \*why\* we did something was intended to reflect our thorough consideration and to fully inform the reader about it. In fact, the only experiment choice that was not completely clarified was the timestep size, and the other reviewer requested clarification for it. We also carefully considered this suggestion, but in the end decided to not make any changes in this regard, especially since the comment was not really negative.*

64. Figure 5: A busy figure trying to show several different things, and with different horizontal axes. (Likewise Fig 9, Fig 10.) Decide to show one thing and show it?

    *Each one of these figures has been split into two figures.*

65. section 3.4: Use "H" for ice thickness. "h" already has its usual meshing meaning of mesh size.

    *Done.*

66. line 431: This casual mention of using $\tau_b$ for mesh adaptation seems to be the first contact with glaciological interests. On page 17.

    *We respond to this in detail under your Recommendation paragraphs (Point 2).*

67. subsection 3.4.1: Speaking as a reader who is making an effort, I am a bit lost at this point on what the difference between global and hybrid is. I see hybrid is better (= closer to uniform ref). Is hybrid more expensive? Generally much of this subsection reads as chain of thoughts of the numerical experimenter, not a purposeful statement of results.

    *Significant changes have been made to improve the flow of this subsection and to clarify the main points within it:*

    - *Changes to the section where we introduce different mesh adaptation algorithms, described in comments 44-48 above, should now more clearly explain the difference between the global and hybrid fixed-point iteration algorithms. We have removed one sentence from the first paragraph that reiterates some of those already-made points.*

- *The final paragraph discusses potential ways to improve the performance of the hybrid algorithm. This may distract readers from the main points of the section and is anyway more suitable for the Conclusions section. In fact, we have already made this point in the Conclusions. So we have removed this paragraph altogether.*

- *Several other sentences were rephrased for clarity, or completely removed. The overall section is therefore decluttered and shorter.*

- *The definitions of diagnostic quantities ($\tilde{e}_H$, $\tilde{e}_{\vec{u}}$, $\|\Delta V_f\|_\infty$, $\|\Delta x_{gl}\|_\infty$) have been moved to the beginning of the Results section, rather than being scattered among different subsections and interwoven with results, as they initially were. We have also typeset all of these in display mode, rather than in inline mode, to increase their visibility.*

- *In the second paragraph we now immediately state that the hybrid algorithm converges twice as fast as the global algorithm.*

- *We have swapped the order of figures, such that adapted meshes are now shown first.*

- *In response to comment 64 above, we have split Fig. 5 into two figures. We believe that the figures are now clearer. We also adapted the text accordingly, such that we first discuss the convergence of the two algorithms. Afterwards, in a separate paragraph, we discuss the temporal distribution of the number of vertices. Initially these two discussions were intertwined.*

68. line 442: "equal observations" means what? Re-word.

    *Modified the sentence for clarity.*

69. subsection 3.4.2: I have already made the point that this physics choice, of which field to adapt upon, should be put much earlier. Needless to say I am not trying to make this point again.

    *We respond to this in detail under your Recommendation paragraphs (Point 2).*

70. equations (13), (14): Also an equation for $\tau_b$, right? (If not I am definitely missing something ... and it is your fault.)

    *We do not consider the difference in $\boldsymbol{\tau}_b$, no. In order to make this clearer, we have moved these equations to the beginning of the Results section to ensure their visibility, but also to immediately clarify how we evaluate the performance of each experiment - \*before\* we present the experiments.*

71. line 500-501: Since basal stress nulls as you cross the grounding line, perhaps intersecting $\tau_b$ and the strain rate magnitude as sensor fields? The latter is telling you about the parts of ice shelves which are interesting. Maybe the Hessian of speed is all you need, not strain rate magnitude (i.e. as sensor field); so intersect $\tau_b$ and $|u|$? The thickness will follow? In any case, the equations of a glacier are (at core) balancing stresses and strain rates to produce smooth velocity and thickness fields.

    *Thank you for the suggestion. We have tried many different choices of sensor fields and many different combinations. In the end, we had to choose a subset of those that were most promising, but also most instructive and digestible to readers.*

    *We have previously experimented with intersecting $\boldsymbol{\tau}_b$ and $\boldsymbol{u}$. These are our observations:*

    - *Since $|\boldsymbol{u}|$ is not differentiable at 0, we first computed metric fields from each of the two velocity components and intersected them together. Similarly for the components of $\boldsymbol{\tau}_b$. Only then do we intersect the two metrics (one from $\boldsymbol{u}$ and one from $\boldsymbol{\tau}_b$). Therefore, four Hessian calculations and three metric intersections had to be done for each of the 200 exported timesteps, at each iteration. This is expensive. In the final section of the paper, and in the final figure, we emphasise that intersecting metric fields from ice velocity and ice thickness is already expensive because it involves multiple Hessian and intersection computations.*

    - *At lower target complexities $\mathcal{C}$, a metric computed from $\boldsymbol{\tau}_b$ always outperforms any other choice of a sensor field. Intersecting it with a metric defined from $\boldsymbol{u}$ is detrimental.*

- *At higher target complexities $\mathcal{C}$, the exact choice of the sensor field is more forgiving, as we already show and discuss in the paper. While the intersection outperforms the $\tau_b$-only metric definition (in the sense of error to number of vertices), this comes at an additional cost (described in the first point above).*

*We have therefore decided not to include this experiment in the paper, since similar observations have already been demonstrated and discussed through the experiments that we present in the paper.*

*However, we must emphasise again that in our work we do not use many of the techniques available to metric-based mesh adaptation. We have explored several of these other techniques too, but, again, in the end we had to make a selection of results to show.*

72. Figure 8: This dark dense figure tells me little and makes me want to look elsewhere. Maybe just reverse the shading.

    *Thanks for the suggestion. We have reversed the shading and agree that the figure is much clearer now.*

73. line 503: Again, "not reflected in Fig 7 to not hinder readability" ... is hindering readability. Since this can be clarified already for Figure 1, I don't think you need to say anything here.

    *Thanks; removed this sentence.*

74. Figure 9 caption: State here that errors $e_h$, $e_u$ are relative to 0.25km uniform ref. The reader jumping to this Figure needs that knowledge first.

    *Done.*

75. lines 534-537: I appreciate this honest assessment of the technique.

    *Thank you.*

76. line 561: "utilized libraries" $->$ "libraries used here"; redundant statement of Firedrake usage

    *Done; removed.*

77. lines 566-568: Mention this fact, that basal shear stress is the preferred sensor field for efficiency (error vs time) in abstract? It is actually a result, not just a "we did some simulations" statement.

    *Done.*

78. line 570: "permille" is not widely used in English literature? But perfectly clear. Interesting.

    *Thanks for pointing this out. It was actually a typo (per mille), so perhaps that is why it stood out. Fixed now.*

79. lines 573-574: suggest replacing "primarily due to the iterature procedure of mesh adaptation algorithms" with "reflecting the cost of our iterative mesh adaptation procedure"

    *Done.*

80. line 575: "utilising" $->$ "using"

    *Done.*

81. line 576: "utilised" $->$ "used"; suggest replacing "features of metric-based adaptation" with "available metric-base adaptation techniques"

    *Thank you; we have modified the sentence to read: "... we have not used several techniques available to metric-based adaptation..."*

82. line 582: "utilised" $->$ "used" ... and etc ... you get the point

    *We have made the modification in this line, as well as in line 585.*

83. line 589: "necessitated" $->$ "required"

    *Done.*

84. lines 597-599: My conclusion has been the opposite, and it is driven by experience switching from idealized experiements with smooth bed topography to modeling whole ice sheets in the BedMachine era. In the later case one has bumpy bed essentially everywhere, and bed topography always locally influences basal shear stress through thermodynamic and liquid water means. Thus everywhere needs about comparable resolution. For example, as we get good bed elevation observations in East Antarctica, then (at least in a 2D sense) everywhere will need good res., with only a slightly enhanced need for resolution near e.g. grounding lines and shear margins. Said a different way, mesh adaptation is effective when part of the domain is easy, and I don't think that is meaningfully true, except for the small area of ice shelves, for real marine ice sheets with real beds. The contrast with mesh adaptation for fluid problems within engineered geometries is strong.

*Thank you for sharing your experience. The other reviewer left a comment along the same lines as this, stating that even though we have showed progress, it is not absolutely convincing that everyone should start using mesh adaptivity. We have thererfore removed this sentence. Furthermore, motivated by this comment, we now end our conclusions with a sentence stating that that "more research is needed to investigate the effectiveness of these methods in larger-scale and less idealised models."*

85. line 600: My favorite line of the paper.

*We are glad that the paper ended on a high note!*

**Anisotropic metric-based mesh adaptation for ice flow modelling in Firedrake**

This document provides a point-by-point response to the comments raised by Reviewer #2.

*Response to Reviewer #2*

*We thank the reviewer for the review! Original comments are written in black, while our responses are provided in blue.*

Dundovic and co-authors present a new ice flow model that represents significant progress in adaptive mesh methods applied to ice flow problems. The key novelty is in an iterative adaption technique that does not simply produce a time-sequence of meshes, each based on the ice sheet state at a given time, revises the mesh at each time step to explicitly obtain optimal solutions at later times. The paper also presents some generally applicable ideas in that type of mesh refinement in plain language (with some illustrative math) and I enjoyed reading those parts in particular. Overall a good paper that should be published and fits GMD well. My comments are minor.

**General Comments**

The paper claims at a couple of points that adaptive mesh (AMR) methods are underused in ice flow modelling and should be used more on the basis of this paper. These are not justified (or technical) points and should be removed (e.g last paragraph of the conclusion). There are several adaptive mesh ice flow models: Ua (often with AMR), BISICLES (always), ISSM (sometimes), UFEMISM, in use on large scale, realistic problems today (the paper references these codes). The model here is tested only on a toy problem with a tiny domain (MISMP+), and even then, halves the domain. There are also well-regarded uniform resolution models (PISM, CISM and others). Clearly, many modellers find AMR not utterly compelling even when it is available. This paper will not change that, because the performance improvements noted relative to fixed-in-time meshes are modest.

*We agree that this was not an accurate assessment, especially since we indeed then go on to cite these various works. We have therefore removed the sentence in the introduction which claims that mesh adaptation methods are underused in glaciology. However, in response to the other reviewer's comment 84, we end our Conclusions section with a sentence stating that "more research is needed to investigate the effectiveness of these methods in larger-scale and less idealised models."*

There is no mention of the expected or observed asymptotic rates of error estimate convergence with $1/N$. Figure 9 shows error estimates, which look to me to be $O(N^{-1})$ in panel (a) and perhaps tending to $O(N^{-2})$ in panel (b). Figure 9 could be modified to show those reference rates, and some discussion made – what is leading order error do you expect from your discretization? The discontinuity Tb (or rather, in dTb/dx in this case, I think) likely reduces you to first order in space. It also sounds from your description (compute the velocity, then advance the thickness) as though you are first order in time.

- *We agree that this should be explicitly mentioned. We have modified the text to do so (at the end of section "Mesh adaptation strategy" and around this figure). However, we have decided not to include the $O(N^{-1})$ and $O(N^{-2})$ convergence rates directly in the figure. The other reviewer has commented (see their Recommendation paragraph) that the figure is too complicated, and has asked for it to be broken down into two separate figures. Since this figure already shows results from five different sets of experiments (one for each choice sensor fields + uniform meshes), we did not want to again complicate it further.*

- *The Euler approximation to the coupled model scheme is indeed first-order in time, as we discuss further in comments 11 and 13 below.*

1. L84 H of $\Omega$ -> H($\Omega$) ?

   *Thank you; this was indeed clumsy. We removed "of $\Omega$" and now simply say: "... spatial discretisation in terms of a mesh, $\mathcal{H}$, which..." Similar change was made in line 122.*

2. L119 $\rho$ and $r_1$ are adequate to describe the mesh parameters, but cannot 'fully support anisotropic mesh adaption' – for that you need the other things you describe (sensor fields etc). Removing the sentence would retain the meaning of the paragraph.

*Thanks for pointing this out. We modified the sentence to now instead read:"Thus the metric-based approach introduces anisotropy in the resulting mesh." as suggested by the other reviewer (comment 22).*

3. Figure 1. I would like to see a zoom of the GL regions – that way it is easier for the reader to see what is meant by the signs of $r_1$ & $r_2$

   *Good idea, thanks. In response to the other reviewer's comments, we expanded this figure to show three different metric fields. On two of the corresponding adapted meshes we zoom in onto the grounding line and on one we zoom in onto the shear margin.*

4. L126: given that the unit size is impossible, why not say 'approximately unit size'?

   *Done.*

5. L134-135. I see what you are saying there, but I think you can say more -in some regions the elements are stretched along x, in other along y.

   *Thank you for the suggestion. We have expanded this sentence to be more specific and have modified the figure to include a zoom-in on the grounding line and shear margin regions of adapted meshes to better demonstrate this variation (as described in comment 3 above).*

6. L147: both h- and p-adaption are 'active and impactful'. Agreed, but citation needed. Also, p-adaption is in many problems the more efficient approach, you might comment on that.

   *We have briefly expanded on the pros and cons of both p- and h-refinement and cited the recent work of Kirby and Shapero, 2024 alongside the already-cited Cuzzone et al., 2018.*

7. L170 : say a bit more about Cst. To me it seems to be space complexity, controlling the mesh density in a 'uniform' fashion in space (crudely, you can obtain a mesh with 2X Cst by splitting each element in two, I know that is not what you actually do). But how is it a time-complexity?

   *We have modified the text for clarity. It now reads: "The space-time complexity provides an estimate for the average number of mesh vertices in the entire mesh sequence $\{\mathcal{H}_i\}_{i=1}^{N_a}$. The number of vertices, however, may vary between meshes, as they are distributed in both space and time among individual meshes in the sequence..." That is to say, $\mathcal{C}_{st}$ is a constraint corresponding to the average number of vertices of the entire mesh sequence in time-dependent mesh adaptation. In contrast, $\mathcal{C}_s$ is a constraint corresponding the number of vertices of a single mesh.*

8. Fig 2 and section 2.5: I did not find Figure 2 helpful – only partly understanding it by reading the (quite wordy) text of section 2.5, rather than the figure illustrating the text. I would like to see these algorithms presented as pseudo-code, which would make section 2.5 more accessible.

   *As suggested by both reviewers, we removed the figure and presented the global and hybrid algorithms as pseudocode. We also tidied up the text as described in comment 47 of the other reviewer (removed duplicated statements, rearranged a few sentences...).*

9. Section 2.6 Interpolation between meshes. Indeed, this is important, at least in principle. It is for the sake of conservation and avoidance of (a type of) numerical diffusion when transferring thickness/temperature between meshes that we use the more restrictive block-structured meshes in BISICLES (which relies on a long-standing AMR library, Chombo, originally developed for shock problems in hyperbolic PDEs, where numerical diffusion is anathema). You might note this.

   *Thank you for sharing this; we included it in the first paragraph of the section.*

10. L318-319 Not quite. Viscous (longitudinal and lateral) stresses matter in at least some parts of ice streams, close to the GL, or where basal traction is low, or in shear margins.

    *Thank you; we added a new sentence that closely follows your explanation. We also cite Greve and Blatter, 2009.*

11. L344 – this is what makes me think your methods is first order in time. Is that the case?

    *The Euler approximation to the coupled model scheme which alternates between solving the diagnostic and prognostic equations is indeed first-order. We have now explicitly stated this.*

12. L349 / eqn 12. Is this really done just to smooth the GL? In which case, you are following the approach of Leguy 2014 as well as AMR. Or is it in fact because eq 12 thought to better represent the physics than the discontinuous laws? The pure 'Weertman' problem (P -> infinity) is harder (Tb is discontinuous across the GL), than the modified form (Tb is continuous across the GL, at least when P is given by the formula in L354* but dTb/dx is not), and the best justification for not addressing it is physics. Also, should $\beta^2$ be $\beta^{-2}$ in eqn 12. So that $Tb = \beta^2 |u|^{1/3}$. * You say that L354is the formula for water pressure under floating ice, but you only need such a formula in the basal traction under grounded ice. Or do you compute P there differently?

- *The motivation to use some form of the Schoof sliding law was indeed just to smooth the GL, as we describe in the text. We already cite Leguy et al., 2014 in the introduction and we have now cited it again here. And just in case the question was referring to why we use a bespoke form of the Schoof sliding law: it is because Firedrake doesn't support hypergeometric functions, which are required for the original Schoof sliding law implementation (Shapero et al., 2021).*
- *Thank you, $\beta^2$ was indeed a typo. Changed to $\beta^{-2}$.*
- *Sentence in L354 related to pressure was indeed confusing. We rewrote the equation to now explicitly consider the grounded and floating parts. We also clarified the text.*

13. L356 'second order' – but only if you supply it with velocity at t + dt/2, rather than t? Seems not to agree with L344.

    *The icepack implementation of the Lax-Wendroff scheme for transport is indeed second-order and can be found here: https://github.com/icepack/icepack/blob/master/src/icepack/solvers/flow_ solver.py#L348. In the code, the second derivative term $\partial^2 h/\partial t^2$ is represented in weak form as a sum of flux contributions in the interior of the domain and at boundaries. However, we have decided to drop "second-order" from this sentence to avoid further confusion. As noted in comment 11, the Euler approximation to the coupled model scheme is indeed first-order.*

14. Fig. 4 / sec 3.2 I am not convinced that you are seeing convergence here. As you note earlier, you need to show a pattern of successively closer solutions, which might apply to the 1km-500m-250 sequence, but you need one more refinement (125m) to look convincing. The gap between 1km and 500 m looks much larger (10x?) than the gap between 500 and 250 m, but once the leading order term in truncation error dominates, you hope to see the gap successively halving (for $O(N^{-1})$) / quartering ($O(N^{-2})$) etc.

    *We completely agree; thank you for pointing this out. We ran a further experiment with the 125m resolution and added the result to this figure, as well as mentioning it in the text. This now (properly) shows that results have indeed sufficiently converged, so the rest of our results remain unaffected.*

15. L368 'The Ice1 experiment starts from *an* initial state'. Just a typo

    *Thank you; we corrected this.*

16. L382 – why this time-step in particular? CFL on your finest mesh?

    *The intention is indeed to satisfy the CFL condition. We now explicitly state this both in the uniform and adaptive mesh experiments. The timestep size $1/24\,a$ is in fact considerably smaller than the CFL condition requires on the finest 125m resolution mesh (approximately $125\,m/1644\,ma^{-1} = 1.824/24\,a$). We took an even smaller timestep to satisfy the CFL condition on adaptive meshes that could have even finer local resolution, since we do not prescribe bounds on minimum element lengths.*

17. L389 – all simulations were run in serial – comment on this, was that just to measure CPU time, or is your method not yet suitable for parallel runs.

    *We modified the sentence in this line and in line 545 (of the unrevised manuscript) to now say that it is possible to run simulations in parallel, but that we indeed ran them in serial for the purposes of CPU time measurements.*

18. Fig 5 captions – define the symbols used on the axes e.g deviation in V.A.F ($||\Delta V_f||_\infty$)

    *Thank you; we have done this in Figure 5 as requested, and also in Figure 4.*

19. Fig 6 (and 7) Zoom in on the GL – as it is I can see there is some difference in your meshes, but not in detail

    *We have done this in both figures.*

20. L 487: 'Ice bends etc' – in SSA? There is no adjusting to buoyant forces in SSA, which assumes hydrostatic balance. The horizontal stretching occurs because all the driving stress from the front / local slopes is resisted by viscous stresses and hence results in horizontal longintudinal strain-rates.

    *Thank you; we simplified this sentence to avoid using the term "hinge zone" and what you pointed out in the comment. Instead we simply say "... on the ice shelf near the grounding line."*

21. L496 'basal stress is discontinuous across the GL' – not according to eqn 12. P -> 0 as you approach the GL from the upstream side (or is P computed differently from the formula in L354?). Also L500: as u->0, P -> $\infty$ eqn 12 becomes $|Tb| = |\beta^2 u^{1/3}|$ - the discontinuity you are seeing at the inflow boundary seems like an arithmetic artefact, with u/|u| being undefined (but also irrelevant since |Tb| = 0)

    *Thanks for pointing this out.*

    - *The basal stress is indeed not discontinuous, so we now instead say that it rapidly diminishes across the grounding line. Similar modification was made in Line 509 of the unrevised manuscript.*
    - *We agree with your assessment of the refinement at the inflow boundary. However, we have decided to drop this sentence for conciseness. It is a minor point in an already quite wordy section.*

22. L575: This paragraph seems spurious.

    *We agree. The main point of this paragraph is in the final sentence, which was further clarified two paragraphs down, so we decided to completely remove this paragraph.*

23. L597: I disagree. You have demonstrated progress, but it is up to the reader to decide whether they should adopt the methods.

    *We have removed this sentence. Detailed response given under the General Comments section.*

**References**

Cuzzone, J. K., Morlighem, M., Larour, E., Schlegel, N., & Seroussi, H. (2018). Implementation of higher-order vertical finite elements in issm v4. 13 for improved ice sheet flow modeling over paleoclimate timescales. *Geoscientific Model Development*, *11*(5), 1683–1694.

Farrell, P. E. (2010). *Galerkin projection of discrete fields via supermesh construction* [Doctoral dissertation, Imperial College London].

Greve, R., & Blatter, H. (2009). *Dynamics of ice sheets and glaciers.* Springer Science & Business Media.

Kirby, R. C., & Shapero, D. (2024). High-order bounds-satisfying approximation of partial differential equations via finite element variational inequalities. *Numerische Mathematik*, 1–21.

Leguy, G., Asay-Davis, X., & Lipscomb, W. (2014). Parameterization of basal friction near grounding lines in a one-dimensional ice sheet model. *The Cryosphere*, *8*(4), 1239–1259.

Shapero, D., Badgeley, J., Hoffmann, A., & Joughin, I. (2021). Icepack: A new glacier flow modeling package in python, version 1.0. *Geoscientific Model Development Discussions*, *2021*, 1–34.

Wallwork, J. G., Knepley, M. G., Barral, N., & Piggott, M. D. (2022). Parallel metric-based mesh adaptation in PETSc using ParMmg. *30^{th} International Meshing Roundtable.* https://doi.org/10.48550/arXiv.2201.02806